# Depth-Width Tradeoffs for Transformers on Graph Tasks

**Gilad Yehudai**[*]
Courant Institute of Mathematical Sciences,
New York University
gy2209@nyu.edu

**Clayton Sanford**[*]
Google Research

**Maya Bechler-Speicher**
Meta AI

**Or Fischer**
Bar-Ilan University

**Ran Gilad-Bachrach**
Department of Bio-Medical Engineering,
Edmond J. Safra Center for Bioinformatics, Tel-Aviv University
Tel Aviv University

**Amir Globerson**
Google Research
Tel-Aviv University

## Abstract

Transformers have revolutionized the field of machine learning. In particular, they can be used to solve complex algorithmic problems, including graph-based tasks. In such algorithmic tasks a key question is what is the minimal size of a transformer that can implement the task. Recent work has begun to explore this problem for graph-based tasks, showing that for sub-linear embedding dimension (i.e., model width) logarithmic depth suffices. However, an open question, which we address here, is what happens if width is allowed to grow linearly, while depth is kept fixed. Here we analyze this setting, and provide the surprising result that with linear width, constant depth suffices for solving a host of graph-based problems. This suggests that a moderate increase in width can allow much shallower models, which are advantageous in terms of inference and train time. For other problems, we show that quadratic width is required. Our results demonstrate the complex and intriguing landscape of transformer implementations of graph-based algorithms. We empirically investigate these trade-offs between the relative powers of depth and width and find tasks where wider models have the same accuracy as deep models, while having much faster train and inference time due to parallelizable hardware.

## 1 Introduction

The transformer architecture [Vaswani, 2017], has emerged as the state-of-the-art neural network architecture across many fields, including language [Brown et al., 2020], computer vision [Dosovitskiy et al., 2021] and molecular analysis [Jumper et al., 2021]. This success prompts a basic question: which algorithms can be implemented using transformers, and how do specific architecture choices affect this capability. Understanding these questions could enable practitioners to design more computationally efficient models without compromising expressivity.

---

[*]Equal contribution

Two significant architectural elements of transformers are the *depth* (the number of layers) and *width* (the dimension of their latent representation). Increasing either property increases the expressivity of the underlying architecture, but the qualitative differences between "shallow and wide" and "deep and narrow" architectures are poorly understood. Given the widespread use of specialized parallel hardware for training and serving modern language models, increasing the width has a limited effect on model latency, provided sufficient memory. A wide range of recent theoretical work [see e.g., Merrill and Sabharwal, 2023b, Liu et al., 2023] explores these fundamental expressivity questions for various benchmark tasks and transformer scaling regimes. However, a complete understanding of the interplay between architecture and algorithmic capabilities—and in particular the *fine-grained* interplay between depth and width—remains elusive. In this work, we aim to crystallize the emergent algorithmic capabilities of transformers as a function of model depth and width, and demonstrate occasions when, contrary to conventional wisdom, the benefits of width eclipse those of depth. Specifically, we focus on the following question:

*What are the algorithmic capabilities obtained by increasing the width of a transformer?*

We ground this question in the context of graph algorithmic tasks, which provide a compelling testbed for understanding transformer reasoning. Graph algorithms serve as a natural "algorithmic playground," encompassing a wide range of well-known problems that span computational classes. Many of these tasks have already been investigated as benchmarks for language models [Fatemi et al., 2023]. Furthermore, previous theoretical works have routinely employed these algorithmic tasks to unveil the capabilities and limitations of graph neural networks [GNNs, Gilmer et al., 2017]. For example, the theoretical analysis of Loukas [2019] revealed lower bounds on the depth and width necessary to determine connectivity. Sanford et al. [2024a] employed similar analysis to elucidate such trade-offs for transformers. However, their results pertain primarily to the powers of depth and a specific edge-list representation. Other scaling regimes—such as transformers with fixed depth and variable width—have yet to be explored in the context of this toolkit.

For such graph reasoning tasks, we are principally concerned with the optimal transformer size scaling as a function of the graph size. Specifically, given a graph with $n$ nodes and an encoding of an algorithmic task, such as determining connectivity or counting the number of triangles, we investigate the width necessary and sufficient to solve the task as a function of $n$. This fixed-depth, variable-width setting is arguably more pertinent given the aforementioned benefits of width in computational efficiency. Indeed, practical applications of transformers to graphs have width that is often much larger than the number of vertices in the graph.[2]

Our theoretical contribution is a novel representational hierarchy that characterizes this depth-width dependence in transformers. Surprisingly, our results show that multiple problems of interest can be solved with fixed depth and linear width with respect to $n$. We provide empirical support for the advantage of these *shallow and wide* models.

Our results elucidate a hierarchy concerning the minimum width needed for constant-depth transformers to represent solutions to various graph algorithmic tasks. Section 5 establishes the relevance of the *linear scaling regime*, where the width of the transformer increases linearly in the number of nodes $n$. Specifically, it considers tasks, such as graph connectivity and detection of fixed length cycles, for which linear width is necessary and sufficient for dense graph input. Section 5 demonstrates that more complex tasks, including subgraph counting and Eulerian cycle verification, require super-linear and near-quadratic width respectively. The resulting hierarchy over transformer widths induced by our collection of positive and negative results is visualized in Figure 1, which ranges from local node-level tasks that can be solved with constant width (such as computing the degree of each node), to arbitrary functions that require quadratic width scaling. Some of the lower bounds are conditional on the well known one vs. two cycles conjecture (see appendix B.1 for a formal statement).

The empirical results of Section 6 demonstrate the relevance of our results in practice. We consider various graph problems, and experiment with varying the depth and width of the model under a constraint on the overall parameter count. Our results demonstrate that shallower and wider models are significantly faster than narrow and deep models.

---

[2]See Table 3 for a list of commonly used graph datasets (including molecular datasets) where the average graph size in the data is less than 40.

## 2 Related Works

**Expressive power of transformers** Transformer with arbitrary depth [Wei et al., 2021, Yun et al., 2020] or arbitrarily many chain-of-thought tokens [Malach, 2023] are known to be universal approximators. Informally, these results suggest that any algorithm can be simulated with a *sufficiently large* transformer. They, however, leave open more fine-grained questions about more compact transformer implementations of a given algorithm, particularly in terms of parameter count and depth.

Various theoretical techniques have been employed to develop a more precise understanding of the relevant scaling trade-offs. For instance, Merrill and Sabharwal [2023a] show that constant-depth transformers with polynomial width can be simulated by $\mathsf{TC}^0$ Boolean circuits, implying that they cannot solve problems like graph connectivity. These results provide essential context for our paper, where we introduce scaling regimes where transformers can and cannot learn such tasks.Similarly, Hao et al. [2022] identify formal language classes that can and cannot be recognized by hard-attention trans-

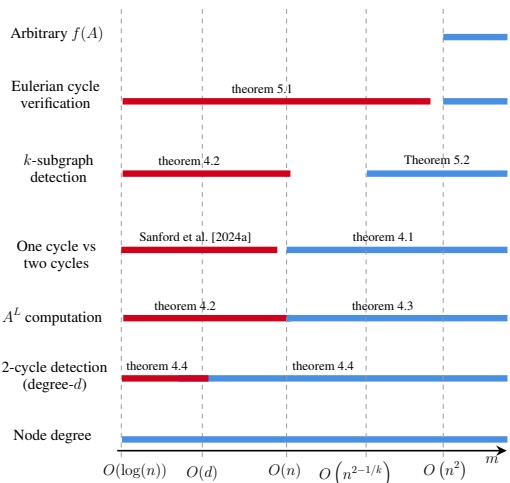

Figure 1: The width complexity hierarchy of graph tasks for transformers with constant depth. Each row visualizes the width regimes where the task is solvable (blue) or hard (red).

formers. Another way to augment transformer expressivity is to use chain-of-thought reasoning [Wei et al., 2022]. Indeed, it has been shown that sufficiently long chains reasoning can simulate finite-state automata [Liu et al., 2023, Li et al., 2024].

A different perspective frames transformers in terms of communication complexity. Sanford et al. [2024c,b] draw an analogy between transformers and the Massively Parallel Computation (MPC) framework [Karloff et al., 2010], similar to prior work linking GNNs to the CONGEST model in distributed computing [Loukas, 2019]. Sanford et al. [2024a] extend this analogy to define a transformer complexity hierarchy for graph tasks.

**Graph transformers and GNNs** Research in transformers and graph neural networks (GNNs) reveals a rich overlap, primarily because both are fundamentally message-passing models. Substantial research effort has been devoted to the study of hybrid architectures to combine GNNs' intrinsic encoding of input graphs and transformers' empirical successes. A variety of approaches to model hybridization have been explored, including the incorporation of graph structure directly into attention layers [Veličković et al., 2018, Brody et al., 2022], as subgraph features in positional encodings [Zhang et al., 2020], and as spectral features of the graph Laplacian matrix [Kreuzer et al., 2021]. The aim of this work is to understand transformer architectural trade-offs rather than to introduce a novel architecture. However, the graph encoding schemes employed in this paper draw upon key insights from this literature. We primarily utilize a simple adjacency encoding in order to isolate the focus of our study to the impacts of changing the embedding dimension, but we also investigate some theoretical and empirical trade-offs for spectral Laplacian-based encodings (Appendix A).

## 3 Problem setting and notations

### 3.1 Transformers

We consider the following setting of transformers: the input is a sequence of $N$ tokens $\mathbf{x}_1, \ldots, \mathbf{x}_N$ where $\mathbf{x}_i \in \mathbb{R}^{d_{\text{in}}}$. We denote by $X^{(0)} \in \mathbb{R}^{d_{\text{in}} \times N}$ the matrix where the $i$-th column is equal to $\mathbf{x}_i$. Each layer of the transformer applies a self-attention mechanism on the inputs, and then an MLP. We denote the input to layer $\ell$ by $X^{(\ell-1)}$. The self-attention at layer $\ell$ with $H$ heads is parameterized by

matrices $K_h^{(\ell)}, Q_h^{(\ell)} \in \mathbb{R}^{m \times m}, V_h^{(\ell)} \in \mathbb{R}^{m \times m}$.[3] It is defined as:

$$Z^{(\ell)} = \sum_{h=1}^{H} V_h^{(\ell)} X^{(\ell-1)} \mathrm{SM}(X^{(\ell-1)^\top} K_h^{(\ell)^\top} Q_h^{(\ell)} X^{(\ell-1)}) ,$$

where SM is row-wise softmax, and $m$ is the hidden embedding dimension. The output is of dimension $Z^{(\ell)} \in \mathbb{R}^{m \times N}$. This is followed by a residual connection, so that the output of the self-attention layer is $\tilde{X}^{(\ell)} = Z^{(\ell)} + X^{(\ell-1)}$.

Finally, we apply an MLP $\mathcal{N}^{(\ell)} : \mathbb{R}^m \to \mathbb{R}^m$ with ReLU activations on each token separately (i.e. each column of $\tilde{X}^{(\ell)}$). The output of the MLP is $X^{(\ell)}$, and is the input to the next layer. We let $\sigma : \mathbb{R} \to \mathbb{R}$ denote the ReLU activation. Critically, we assume that the MLP layer can compute arbitrary functions of each individual token embedding $X_i^{(\ell)}$. While involved, this assumption is common in the theoretical literature [see e.g. Sanford et al., 2024b] and enables a theoretical understanding of the modeling limitations imposed by the attention layer. However, we note that in all our constructions, the size of the MLP is at most polynomial in the number of nodes in the input graph. Our MLP layers further incorporate arbitrary positional encodings of the index $i$ of each token embedding $\tilde{X}_i^{(\ell)}$, i.e. $\mathcal{N}^\ell(\tilde{X}^{(\ell)}) = (g(\tilde{X}_1^{(\ell)}, 1), \ldots, g(\tilde{X}_N^{(\ell)}, N))$, for some $g$. This formalism allows standard transformer positional encodings to be implemented in our theoretical model. Normalization layers—which are not explicitly accounted for in our transformer definition—can be similarly incorporated by taking advantage of the arbitrary MLP assumption without changing the results.

The bit-precision of all our transformers will be $O(\log n)$, where $n$ is the number of input tokens. This is a common assumption in many previous works Sanford et al. [2024c,b], Merrill and Sabharwal [2023b, 2024], and is relatively mild. It is also a necessary requirement for representing a number of size $n$, e.g. when having positional embeddings for $n$ tokens. We will denote $m = \max(m_0, \ldots, m_L)$ the embedding dimension, and $d_{\mathrm{in}}$ the input dimension.

**Graph Inputs**   Because the network topology of transformers (unlike GNNs) does not encode the structure of an input graph, graph structure must be encoded explicitly into the input of the transformer. Choosing the format of this *tokenization scheme* for graph inputs is a significant modeling decision. The most fundamental aspect of that choice involves deciding between edge-wise and node-wise schemes. Sanford et al. [2024a] employ an edge-list tokenization that converts the graph into a sequence of discrete edge tokens. However, this scheme suffers computationally for large graphs; the quadratic computational bottleneck of self-attention can result in an $\Omega(n^4)$ runtime for dense graphs with $n$ nodes.

In contrast, we consider node-wise encodings, where each node corresponds to exactly one input embedding. Our primary theoretical results use what is arguably the simplest such representation: the *node-adjacency tokenization scheme*. In this representation, each input embedding directly encodes a node's edges as an adjacency vector. That is, for graph $G$ with $n$ nodes and adjacency matrix $A \in \{0,1\}^{n \times n}$, the $i$th token input $\mathbf{x}_i \in \mathbb{R}^n$ to the transformer is defined as the $i$th row of $A$. An alternative node-wise scheme is the Laplacian eigenvector tokenization [Kreuzer et al., 2021], which captures global graph structure node-wise in a lossy manner with the most significant eigenvectors of the Laplacian matrix. We empirically contrast the node-adjacency scheme with both alternatives in Section 6, and we further explore the representational properties of the Laplacian tokenization in Appendix A.

The adjacency node embedding is not permutation invariant, similar to the edge-list embedding used in Sanford et al. [2024a]. Namely, if the tokens contain positional embeddings, then changing the order of the tokens may also change the output. These embeddings are used for technical convenience, especially since spectral embeddings that are permutation invariant are difficult to analyze for combinatorial problems (such as connectivity, subgraph counting, etc.). In section 6 we provide experiments that justify the use of such embedding on real-world datasets.

---

[3]Except for the first layer where $K_h^{(\ell)}, Q_h^{(\ell)} \in \mathbb{R}^{d_{\mathrm{in}} \times d_{\mathrm{in}}}, V_h^{(\ell)} \in \mathbb{R}^{m \times d_{\mathrm{in}}}$

# 4 The Expressive Power of a Linear Embedding Dimension

In this section we focus on two graph reasoning tasks: the $1$ vs. $2$ cycle problem and the cycle detection problem. We show that a fixed-depth transformer can solve these tasks with only linear width. We also show that these results are tight. All the proofs for the theorems in this section are in Appendix B. Finally, in Sec. 4.3 we show that for cycle detection in bounded degree graphs, sublinear width suffices. Note that the node degree task as appears in fig. 1 (namely, calculating the degree of each node) can be easily solved with an $O(\log(n))$ embedding size, as shown e.g., in Sanford et al. [2024a]

## 4.1 A linear width solution for $1$ vs. $2$ cycle detection

Previous works on the connection between the MPC model and transformers have shown conditional lower bounds for solving certain tasks. One such lower bound includes the $1$ **vs.** $2$ **cycle** problem. Namely, determine whether an $n$-node graph is one cycle of length $n$ or two cycles of length $\frac{n}{2}$. Conjecture 13 from Sanford et al. [2024a] (see also Ghaffari et al. [2019]) states that any MPC protocol with $n^{1-\epsilon}$ memory per machine for any $\epsilon \in (0, 1)$ cannot distinguish between the two cases, unless the number of MPC rounds is $\Omega(\log n)$. This condition implies that transformers with an embedding dimension of $O(n^{1-\epsilon})$ cannot solve this task. For a formal definition, see appendix B.1. We demonstrate the tightness of this bound by showing that a transformer with linear embedding dimension *can* solve this task:

**Theorem 4.1.** *There exists a transformer with $2$ layers of self-attention, and embedding dimension $O(n)$ that solves the $1$ vs. $2$ cycle problem.*

The proof intuition is that the graph contains only $n$ edges, we can stack all of them into a single token. After doing so, we can use the MLP to solve it. This demonstrates the brittleness of communication-based transformer lower bounds, since even a slight increase in the embedding dimension breaks them. With that said, this result is not surprising, since the graph in this task is very sparse.

The connectivity problem for general graphs cannot be solved in this manner. The reason is that for graphs with $n$ nodes there are possibly $\Omega(n^2)$ edges, and thus it is not possible to compress the entire graph into a single token with linear embedding dimension. However, the connectivity problem can still be solved by increasing the embedding dimension by polylog terms, as we explain below.

Ahn et al. [2012] uses linear sketching to solve the connectivity problem on general graphs with $O(n \log^3(n))$ total memory. The idea is to use linear sketching, which is a linear projection of the adjacency rows into vectors of dimension $O(\text{poly} \log(n))$. Although this is a lossy compression, it still allows to solve the connectivity problem. As a consequence, we can use a similar construction as in Theorem 4.1 where we first apply the sketching transformation to each token (i.e. row of the adjacency), then embed all the tokens, into a single token and use the MLP to solve the problem using the algorithm from Ahn et al. [2012] (Theorem 3.1). Note that this construction works only in high probability rather than deterministically, since the linear sketching requires using a random matrix, which compresses successfully the adjacency matrix only with some high probability. Thus, we don't present it here formally, but rather as a proof sketch.

Our next results include problems that cannot be solved by simply compressing all the information of the graph into a single token, and require a more intricate use of the self-attention layers.

## 4.2 Linear width is necessary and sufficient for $2$-cycle detection

This section shows that for the problem of cycle detection, linear width is sufficient for solving the task with fixed-depth, and is also necessary. We consider the 2-cycle detection problem for directed graphs. The task is to find nodes $u$ and $v$ such that the edges $(u, v)$ and $(v, u)$ exist. We begin with the lower bound for this case.

**Theorem 4.2.** *Let $T$ be a transformer with embedding dimension $m$ depth $L$, bit-precision $p$ and $H$ attention heads in each layer. Also, assume that the input graphs to $T$ are embedded such that each token is equal to a row of the adjacency matrix. Then, if $T$ can detect $2$-cycles on directed graphs, the following must hold: (1) If $T$ has residual connections then $mpHL = \Omega(n)$; or (2) If $T$ doesn't have residual connections then $mpH = \Omega(n)$.*

The proof uses a communication complexity argument, and specifically a reduction to the set disjointness problem. For a formal definition see Appendix B. This lower bound is stronger than the lower bounds in Section 4.1 in the sense that they cannot be circumvented by logarithmic depth. For example, assume that the embedding dimension is $m = O(n^{1-\epsilon})$ for some $\epsilon \in (0, 1)$, and that $p, H = O(\log(n))$ (which is often the case in practice). Then, transformers with residual connection require a depth of $\Omega(n^\epsilon)$ to solve this task, which is beyond logarithmic, while transformers without residual connections cannot solve it. This lower bound is also unconditional, compared to the lower bounds from Sanford et al. [2024a] which are conditional on the hardness of the 1 vs. 2 cycle problem (Conjecture 13 therein). The caveat of this lower bound is that it relies on having the input graph specifically embedded as rows of the adjacency matrix.

We now turn to show an upper bound for this task. We will show an even more general claim than detecting 2-cycles. Namely, that depth $L$ transformers with $\Omega(n)$ embedding dimension can calculate the $L$-th power of an adjacency matrix of graphs. In particular, the trace of $A^L$ equals the number of cycles of size $L$ in the graph (multiplied by an appropriate constant). Thus, a 2-layer transformer with linear embedding dimension can already solve the directed 2-cycle problem.

**Theorem 4.3.** *There exists an $O(L)$-layer transformer with embedding dimension $m = O(n)$ such that, for any graph embedded as rows of an adjacency matrix $A$, the output of the transformer in the $i$-th token is the $i$-th row of $A^L$.*

The constructive proof of this theorem carefully selects key and query parameters to ensure that the output of the softmax matrix approximately equals $A$. This enables an inductive argument that encodes $A^\ell$ in the value matrix of the $\ell$th layer, in order to compute $A^{\ell+1}$.

The above theorem shows that having a transformer with embedding dimension proportional to the graph size is representationally powerful. Namely, $A_{i,j}^L$ counts the number of walks of length $L$ form node $i$ to node $j$. This also allows to determine whether a graph is connected, by checking whether $A^n$ doesn't contain any zero entries. Although, there are other algorithm (e.g. those presented in Section 4.1) that can solve the connectivity task more efficiently.

## 4.3 A sublinear width solution for cycle detection in bounded degree graphs

The results of the previous section establish that *for worst-case graph instance*, transformers with node-adjacency tokenizations require a linear embedding dimension to solve simple graph reasoning tasks, such as 2-cycle detection. This fact is perhaps unsurprising because each node-adjacency tokenization input is a length-$n$ boolean vector, which must be compressed in a lossy manner if the embedding dimension $m$ and bit precision $p$ satisfy $mp = o(n)$. It is natural to ask whether such results apply to sparse graphs as well, such as graphs with bounded degree. Here, we show that requisite embedding dimension to detect 2-cycles scales linearly with the degree of the graph.

**Theorem 4.4.** *For any $n \in \mathbb{N}$ and $d \leq n$, there exists a single-layer transformer with embedding dimension $O(d \log n)$ that detects 2-cycles in any graph with node degree at most $d$. This embedding dimension is optimal up to logarithmic factors.*

The proof reduces the dimensionality of the input adjacency matrix by incorporating vector embeddings from Sanford et al. [2024c] into the key, query, and value matrices to produce a "sparse attention unit," whose activations are large when a respective cycle exists.

## 5 The expressive power of sub-quadratic embedding dimension

Thus far, we showed that there are graph problems that can be solved with fixed depth and linear width. A natural question is if there are problems where larger width is necessary? Intuitively, quadratic width should suffice for solving any task, since it can be used to record the entire graph, and then a sufficiently expressive MLP can solve the task. But are there problems where quadratic width is necessary. In Sec. 5.1 we show that for the problem of Eulerian cycle detection, quadratic width is necessary. This leads to the interesting question of whether there are problems that require super-linear but sub-quadratic width (i.e., between linear and quadratic). We offer a first result in this direction by showing in Sec. 5.2 that such width is sufficient for the problem of sub-graph counting.

### 5.1 Necessity of nearly-quadratic width for Eulerian cycle detection

As mentioned above, quadratic width trivially suffices for solving graph problems. It is therefore interesting to understand whether this upper bound is tight for some graph problems. We affirmatively answer this question with the *Eulerian cycle verification* problem. Given a graph and a list of "path fragments," where each fragment consists of a pair of subsequent edges in a path, the goal of the Eulerian cycle verification problem is to determine whether the properly ordered fragments comprise an Eulerian cycle[4]. We establish the hardness of this problem on multi-graphs under the well-accepted "1 vs. 2 cycle" conjecture.

**Theorem 5.1.** *Under Conjecture 2.4 from Sanford et al. [2024b], the Eulerian cycle verification problem on multigraphs with self loops cannot be solved by transformers with adjacency matrix inputs if $m = O\left(n^{2-\epsilon}\right)$ for any constant $\epsilon > 0$, unless $L = \Omega(\log(n))$.*

The proof relies on a novel reduction to the 1 vs. 2 cycle problem. The difficulty with the reduction is that the one vs. two cycles problem consists of a sparse graph with only $n$ edges and nodes, and to show a sub-quadratic lower bound we need a dense graph with $\Omega(n^2)$ edges and $n$ nodes. To this end, we do a random projection of an instance of the one vs. two cycles problem with $n^2$ nodes to a multigraph with $n$ nodes, where each node represents $n$ nodes in the original graph, but keeping all the edges. This allows us to produce a dense multigraph, for which solving the Eulerian cycle verification problem will solve the one vs. two cycles problem in the original graph, before the projection.

### 5.2 Sub-quadratic solutions to sub-graph counting

Subgraph detection and counting are central tasks for fields as diverse as biology, organic chemistry, and graph kernels (see Jin et al. [2020], Jiang et al. [2010], Pope et al. [2018], Shervashidze et al. [2009]; see also the discussion in Chen et al. [2020]). In what follows we show that this problem can be solved with transformers with fixed depth, but width that is between linear and quadratic.

**Theorem 5.2.** *Let $k, n \in \mathbb{N}$, and let $G'$ be a graph with $k$ nodes. There exists a transformer with $O(1)$ self-attention layers and embedding dimension $O\left(n^{2-1/k}\right)$ that, for any graph $G$ of size $n$, counts the number of occurrences of $G'$ as a subgraph of $G$.*

In contrast, known GNN constructions [e.g., see Chen et al., 2020] entail higher-order constructions with $k$-IGN models, which require computing practically-infeasible $k$-order tensors. These models require a depth of $k$ to recognize subgraph of size $k$, where each layer uses a $k$-tensor instead of standard matrices, which requires $n^k$ parameters each. Even for $k = 3$ or $k = 4$ this may be too large for graphs with a few hundreds of nodes.

Our proof implements the seminal "tri-tri-again" algorithm [Dolev et al., 2012] using transformers. Given a graph with $n$ nodes, we partition the nodes into $n^{1/k}$ disjoint sets, each containing $n^{1-1/k}$ nodes. For each possible combination of $k$ such sets, we use an MLP to count the given subgraph in it. There are $\binom{n^{1/k}}{k} \leq n^{k \cdot 1/k} = n$ such combinations of sets, each one containing at most $n^{2-2/k}$ edges. Thus, each token with a large enough embedding dimension can simulate one combination of subsets, cumulating all the relevant edges. Note that subgraph detection is a sub-task of counting, where the number of occurrences is larger than $0$.

Theorem 5.2 can be compared to Theorem 23 in Sanford et al. [2024a] that provides a construction for counting $k$-cliques using transformers with sub-linear memory and additional blank tokens. There, it was shown that it is possible to count $k$-cliques with a transformer of depth $O(\log \log(n))$, however the number of blank tokens is $O(n^{k-1})$ in the worse case. Here, *blank tokens* refer to empty tokens that are appended to the input and used for scratch space as defined in Sanford et al. [2024a]. In our result, we require a depth of $O(1)$, and the total number of tokens is $n$, while the embedding dimension is super-linear (but sub-quadratic). Thus, our solution has better memory usage, since the increase in width is only polynomial, but the number of tokens is $n$ instead of $O(n^k)$.

**Remark 5.3.** *We note that the overall computation time of our depth $O(1)$ model above is still exponential in $k$ due to the size of the MLP. This is also the case in Theorem 23 from Sanford et al. [2024a]. In fact, it is a common conjecture in the literature that no (possibly randomized) algorithm*

---

[4]Recall that a Eulerian cycle is a cycle over the entire graph that uses each edge exactly once.

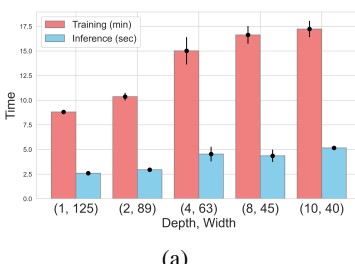 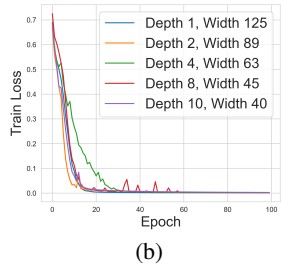 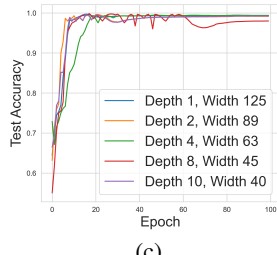

| (a) | (b) | (c) |

Figure 2: Training and inference times (a), training loss curves (b), and accuracy curves (c) for the connectivity task over graphs with $100$ nodes, across transformers with approximately 100k parameters, varying in width and depth.
While the loss and accuracies remain consistent, shallow and wide transformers demonstrate significantly faster training and inference times.

*can detect $k$-cliques in time less than $O(n^{2k/3-\epsilon})$ for any $\epsilon > 0$ (see Hypothesis 6 in Williams [2018]).*

# 6 Experiments

We showcase the effect of width growth in transformers by training a variety of models on synthetic graph algorithmic tasks. Section 6.1 establishes the relative advantages of scaling width over depth by training a family of models with similar parameter counts and variable network topologies; our results show that the shallower and wider models yield the same accuracy as the deeper models, but with faster training and inference. Section 6.2 evaluates the *critical width* at which the substructure counting task becomes solvable, and shows that it is roughly linear. Section 6.3 demonstrates the trade-offs of each graph encoding scheme (node-adjacency versus edge-list versus Laplacian-eigenvector) on different tasks.[5]

In our experiments, we used a standard transformer architecture using Pytorch's transformer encoder layers [Paszke et al., 2019]. Specifically, each layer is composed of Multi-Head Self-Attention, Feedforward Neural Network, Layer Normalization and Residual Connections. More experimental details are provided in Appendix E.

For all experiments in Section 6.1 and Section 6.2, we used the adjacency rows tokenization as described in Section 5.2. Details of the implementation are described in Appendix D. We considered the tasks of connectivity, triangles count, and 4-cycle count. For the counting tasks, we used the substructure counting dataset from Chen et al. [2020], where each graph was labeled with the number of pre-defined substructures it contains, as a graph regression task. For the connectivity tasks, we generated synthetic graphs, and the label indicates whether the graph is connected or not. All the datasets information is described in detail in Appendix E.

## 6.1 Empirical trade-offs between width and depth

In this subsection, we examine the empirical trade-offs between depth and width in transformers. We show that when using transformers that are shallow and wide, the training and inference times are significantly lower than when using deep and narrow transformers. This is while test error and training convergence rates are empirically similar.

We trained a transformer with a fixed amount of 100k parameters split between varying depth and width. We examine how the running time, loss, and generalization depend on the width and depth. We examine the following pairs of (depth, width): $(1, 125), (2, 89), (4, 63), (8, 45), (10, 40)$. We train each model for $100$ epochs and examine the following: the total training time, the total inference time, the training loss and test performance (accuracy for classification and MAE for regression). We

---

[5]Code is provided in the Supplementary Material.

repeat this experiment with graph sizes 50 and 100. We report the averages over 3 runs with random seeds. The hyper-parameters we tuned are provided in Appendix E.

The results for the connectivity task over 100 nodes are presented in Figure 2. Additional results for the counting tasks and different graph sizes, present the same trends, and are provided in Appendix D due to space limitations. As shown in Figure 2(b) and Figure 2(c), the training loss and accuracy remain consistent across all depth and width configurations. However, Figure 2(a) reveals that shallow and wide transformers significantly reduce the total training and inference time compared to their deeper and narrower counterparts. This may be due to the ability of GPUs to parallelize the computations across the width of the same layer, but not across different layers.

## 6.2 Minimum width for substructure counting

In this subsection, we show that the transformer width required to fit the substructure counting tasks increases sub-quadratically with the number of nodes, as argued in Section 5.2. The experiments considered graphs with increasing numbers of nodes, ranging from 50 to 400 in increments of 50, and transformer widths varying from 100 to 800 in increments of 100. For each combination of graph size and transformer width, we determined the critical width at which the model failed to fit the data. The *critical width* is defined as the width where the training loss plateaued at more than 0.05. To determine the critical width, we conducted a grid search over each combination of graph size and model width, and selected the model that fitted the data best. The hyper-parameters we considered are provided in Appendix E.

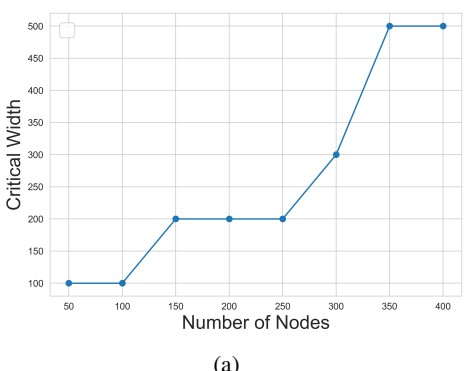

(a)

Figure 3: Critical width evaluation for the 4-Cycle Count task, using a transformer with 1 layer. The points indicate the critical width at which the model fails to fit the data.

To isolate the effect of width, we used a transformer model with one layer. We used two attention heads to ensure there exists a width for which the model can fit the data for all the evaluated graph sizes. The results for the 4-Cycle count task are presented in Figure 3. Due to space limitations, the results for the triangle count task are deferred to Appendix D. Figure 3 shows that the critical width increases roughly linearly with the graph size.

## 6.3 Comparison of graph encodings

In this paper, we focus on a graph tokenization where each row of the graph's adjacency matrix is treated as a token for the model. This tokenization offers significant efficiency advantages for dense graphs, as the edge-list representation requires $O(n^2)$ tokens, whereas the adjacency-row representation reduces this to $O(n)$. To validate the effectiveness of this tokenization approach in practice, we evaluate it on real graph datasets.

We compared the adjacency-row representation to the edge-list representation by training a transformer model on three Open Graph Benchmark (OGB) Hu et al. [2020] datasets: **ogbg-molhiv**, **ogbg-molbbbp**, and **ogbg-molbace**. In ogbg-molhiv, the task is to predict whether a molecule inhibits HIV replication, a binary classification task based on molecular graphs with atom-level features and bond-level edge features. ogbg-bbbp involves predicting blood-brain barrier permeability, a crucial property for drug development. ogbg-bace focuses on predicting the ability of a molecule to bind to the BACE1 enzyme, associated with Alzheimer's disease. We also evaluated a tokenization using the eigenvectors and eigenvalues of the graph Laplacian, as commonly used in the literature [Dwivedi and Bresson, 2021, Kreuzer et al., 2021]. More experimental details, including the dataset statistics can be found in Appendix E.

The results of our evaluation are summarized in Table 1. In all three tasks, the adj-rows representation achieved better ROC-AUC scores than the edge-list representation. In two out of the three, it also

Table 1: ROC-AUC performance metrics for different graph representations: Edge List, Adjacency Rows, and Laplacian Eigenvectors (LE), averaged over 3 random seeds.

| Tokenization | Dataset | | |
|---|---|---|---|
| | MOLHIV | MOLBBBP | MOLBECA |
| **EdgeList** | $54.01_{\pm 1.38}$ | $64.73_{\pm 1.66}$ | $66.06_{\pm 3.89}$ |
| **AdjRows** | $61.87_{\pm 1.10}$ | $67.63_{\pm 2.57}$ | $68.64_{\pm 2.34}$ |
| **LE** | $68.11_{\pm 1.52}$ | $55.31_{\pm 4.79}$ | $63.61_{\pm 2.31}$ |

improved upon the commonly used Laplacian eigenvectors representations. The results suggest that the adjacency representation we use in this paper is empirically effective, and should be considered alongside the commonly used Laplacian eigenvector representation.

# 7   Discussion and Future Work

This paper uses a collection of graph algorithmic tasks—including subgraph detection, one vs two cycles, and Eulerian cycle verification—to demonstrate the powers of width bounded-depth transformers that take as input node adjacency encodings. These results include sharp theoretical thresholds that demonstrate the trade-offs between constant, linear, quadratic, and intermediate width regimes. Our empirical results validate the efficiency and accuracy of our choice of scaling regime and embedding strategy.

There are numerous possible extensions of this work. One future direction is to study different graph tokenization schemes, beyond the node-adjacency encoding of this work and edge-list encoding of Sanford et al. [2024a]. A particularly notable alternative is the smallest eigenvectors of the graph Laplacian, presented as a vector of components for each node. This spectral embedding is a standard embedding scheme for GNNs, and the techniques and tasks developed in this paper would likely be relevant to proving similar bounds. We provide a preliminary exploration of the trade-offs between the intrinsically local characteristics of adjacency-based tokenization schemes and more global spectral approaches in Appendix A. Another future direction is to study the optimization and generalization capabilities of transformers to solve graph problems, beyond the expressiveness results presented in this work.

## Acknowledgments

This work was supported in part by the Tel Aviv University Center for AI and Data Science (TAD), the Israeli Science Foundation grants 1186/18 and 1437/22. OF was partially supported by the Israeli Science Foundation (grant No. 1042/22 and 800/22). We thank Joan Bruna for the helpful discussions while this work was being completed.

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

# A   Alternative tokenization approaches

While the primary aim of this paper is to study the properties of the node-adjacency tokenization scheme in terms of its width and depth trade-offs, we also establish clear trade-offs between this scheme and other encoding schemes. The *Laplcian-eigenvector tokenization* passes as input to the transformer each node's components of the most significant eigenvectors.

Concretely, let $A \in \mathbb{R}^{n \times n}$ be the adjacency matrix of a graph and $D$ the diagonal degree metrix. The *Laplacian matrix* is defined as $\mathcal{L} = D - A$. Denote the eigenvectors of $\mathcal{L}$ as $\mathbf{v}_1, \ldots, \mathbf{v}_n$ with respective eigenvalues $0 = \lambda_1 \leq \cdots \leq \lambda_n$. For some embedding dimension $m$, we let the $m$-dimensional Laplacian-eigenvector tokenization be $\mathbf{y}_1, \ldots \mathbf{y}_n$, where $\mathbf{y}_i = (\mathbf{v}_{1,i}, \ldots, \mathbf{v}_{m,i})$; we encode the eigenvalues as well as $\mathbf{y}_0 = (\lambda_1, \ldots, \lambda_m)$. We contrast this with node-adjacency encodings of embedding dimension $m$, whose $i$th input is $\mathbf{x}_i = \phi(A_i)$.

We note several illustrative toy tasks that demonstrate trade-offs between the two graph tokenization schemes.

**Node-adjacency advantage at local tasks**   The node-adjacency tokenization is amenable for analyzing local structures around each node. Most simply, the degree of each node can be computed in a sequence-wise manner with node-adjacency tokenization with embedding dimension $m = 1$ by simply computing the inner products $\langle \mathbf{1}_n, \mathbf{x}_i \rangle$. Constructions like Theorems 4.3 and 4.4 further demonstrate the abilities of adjacency encodings to aggregate local structures.

In contrast, choosing the smallest eigenvectors in the alternative encoding makes it impossible to even compute each node degree without having embedding dimension $m$ growing linearly in the node count $n$[6].

**Laplacian-eigenvector advantage at global tasks**   In contrast, the most significant graph Laplacian provide high-level information about the global structure of the graph. Most notably, the tokenization trivializes the connectivity task because a graph is disconnected if and only if its second-smallest eigenvalue is zero; transformers with the node-adjacency tokenization require either depth $\Omega(\log n)$ or width $\Omega(n)$ to solve the same problem.

Other properties of structured graphs reveal themselves with low-dimensional Laplacian-eigenvector tokenizations. For instance, the relative position of a node in a lattice or ring graph are encoded in the most significant eigenvectors. Graph clustering algorithms could be inferred by transformers that take spectral encodings as input and simulate algorithms like $k$-means. The hardness of graph connectivity with the adjacency encoding translates to hardness results for efficiently simulating clustering algorithms.

**Quadratic embedding equivalence**   Critically, the above trade-offs occur in small embedding dimensions. In the regime where $m = \Omega(n^2)$ and MLPs are universal approximators, both tokenization schemes are universal. The entire graph can be encoded in a single token, which can then convert between $A$ and the spectrum of $\mathcal{L}$.

# B   Proofs from section 4

## B.1   The $1$ vs. $2$ cycle conjecture

The most notable conjecture in distributed computing is the 1 vs. 2 cycle conjecture, which is a common method for providing conditional lower bounds in the MPC model. See Definition 2 in Sanford et al. [2024a] for a $(\gamma, \delta)$-MPC protocol. In simple words, this is a distributed computing protocol, where the input has length $n$, which is distributed to $\Theta(n^{1+\gamma-\delta})$ machines, each having

---

[6]Consider the task of computing the degree of a particular node of a graph consisting of $\frac{n}{3}$ disconnected linear subgraphs, each with three nodes connected by two edges. The zero eigenvalue thus has multiplicity $\frac{n}{3}$, and hence the eigenvectors $\mathbf{v}_1, \ldots, \mathbf{v}_{n/3}$ exist solely as indicators of connected components. Therefore, if nodes $i, j, k$ comprise a cluster and $m \leq \frac{n}{3}$, then their embeddings $\mathbf{y}_i, \mathbf{y}_j, \mathbf{y}_k$ are identical.

memory of $s := \Theta(n^\delta)$. Each communication round, each machine calculates an arbitrary local message bounded by size $s$, and all the messages are sent simultaneously. The 1 vs. 2 cycle conjecture is the following:

**Conjecture B.1** (Conjecture 13 in Sanford et al. [2024a]). *For any $\gamma > 0$ and $\delta \in (0, 1)$, any $(\gamma, \delta)$-MPC protocol that distinguishes a single length-$n$ cycle from two disjoint length-$n/2$ uses $\Omega(n)$ communication rounds.*

## B.2 Proof of Theorem 4.1

**Theorem 4.1.** *There exists a transformer with $2$ layers of self-attention, and embedding dimension $O(n)$ that solves the 1 vs. 2 cycle problem.*

*Proof.* The proof idea is to embed all the information about the graph into a single token, and then offload the main bulk of the solution to the MLP. For that, the first layer will transform the input of each node from adjacency rows to only indicate its two neighbors. The second layer will embed all the information over the entire graph into a single token.

We now define the construction of the transformer. The input to the transformer are adjacency rows, where we concatenate positional encodings that include the row number. Namely, the $i$-th input token is equal to $\begin{pmatrix} \mathbf{x}_i \\ i \end{pmatrix}$, where $\mathbf{x}_i$ is the $i$-th row of the adjacency matrix of the graph. The first layer of self-attention will not effect the inputs. This can be thought of as choosing $V = \mathbf{0}$ (while $K$ and $Q$ are arbitrary), and using the residual connection so that the tokens remain the same as the input. We now use Lemma B.2 to construct a 3-layer MLP that changes the embedding of each token such that it includes for each node its neighbors. The MLP does not change the positional encoding, this can be done since ReLU networks can simulate the identity function by $z \mapsto \sigma(z) - \sigma(-z)$, where $\sigma$ represents the ReLU function. We add another layer to the MLP that maps $\mathbb{R}^3 \ni \mathbf{v}_i \mapsto \mathbf{u}_i \in \mathbb{R}^{3n}$, where $(\mathbf{u}_i)_{3(i-1)+1:3i} = \mathbf{v}_i$ and all the other entries of $\mathbf{u}_i$ are equal to $0$.

The second layer of self-attention will have the following matrices: $K = Q = \mathbf{0}_{3n \times 3n}, V = n \cdot I_{3n}$. Since we used the zero attentions, all the tokens attend in the same way and using the exact same weight to all other tokens. The Softmax will normalize the output by the number of tokens, namely by $n$. Hence, after applying the $V$ matrix, all the output tokens of the second layer of self-attention will be equal to the sum of all the tokens that were inputted to the second layer.

In total, we get that the output of the second layer of attention is a vector with $3n$ coordinates, where each 3 coordinates of the form $\begin{pmatrix} i \\ j \\ k \end{pmatrix}$ represent the two edge $(i, k), (j, k)$. Thus, the entire information of the graph is embedded in this vector.

Finally, we use the MLP to determine whether the input graph, whose edges are embedded as a list of edges, is connected. This can be done by an MLP since it has the universal approximation property Cybenko [1989], Leshno et al. [1993]. Although we don't specify the exact size of this MLP, it can be bounded since there are efficient deterministic algorithms for determining connectivity. These algorithms can be simulated using ReLU networks.

Note that the output of the connectivity problem is either $0$ or $1$, thus it is enough to approximate a solution of this task up to a constant error (say, of $\frac{1}{4}$), and then use another layer to threshold over the answer.

$\square$

**Lemma B.2.** *There exists a 3-layer MLP $\mathcal{N} : \mathbb{R}^n \to \mathbb{R}^2$ such that for every vector $\mathbf{v}$ where there are $i, j \in [n]$ with $(\mathbf{v})_i = (\mathbf{v})_j = 1$ and $(\mathbf{v})_k = 0$ for every other entry we have that either $\mathcal{N}(\mathbf{v}) = \begin{pmatrix} i \\ j \end{pmatrix}$ or $\mathcal{N}(\mathbf{v}) = \begin{pmatrix} j \\ i \end{pmatrix}$.*

*Proof.* The first layer of the MLP will implement the following function:

$$\mathbf{v} \mapsto \begin{pmatrix} \sum_{i=1}^n i \cdot \mathbb{1}((\mathbf{v})_i = 1) \\ \sum_{i=1}^n i^2 \cdot \mathbb{1}((\mathbf{v})_i = 1) \end{pmatrix} \ .$$

This is a linear combination of indicators, where each indicator can be implemented by the function $f(z) = \sigma(x) - \sigma(x - 1)$. Note that the input to the MLP is either 0 or 1 in each coordinate, thus the output of this function will be $\begin{pmatrix} i + j \\ i^2 + j^2 \end{pmatrix}$. We have that $i + j$ and $i^2 + j^2$ determine the values of $i$ and $j$. This means that if $i_1 + j_1 = i_2 + j_2$ and $i_1^2 + j_1^2 = i_2^2 + j_2^2$ then $i_1 = i_2$ or $i_1 = j_2$ and similarly for $i_2$. Since there are $O(n^2)$ different possible values, we can construct a network with 2-layers and $O(n^2)$ width that outputs $\begin{pmatrix} i \\ j \end{pmatrix}$ up to changing the order of $i$ and $j$.

$\square$

### B.3  Proof of Theorem 4.2

**Theorem 4.2.** *Let $T$ be a transformer with embedding dimension $m$ depth $L$, bit-precision $p$ and $H$ attention heads in each layer. Also, assume that the input graphs to $T$ are embedded such that each token is equal to a row of the adjacency matrix. Then, if $T$ can detect 2-cycles on directed graphs, the following must hold: (1) If $T$ has residual connections then $mpHL = \Omega(n)$; or (2) If $T$ doesn't have residual connections then $mpH = \Omega(n)$.*

*Proof.* Our proof relies on a communication complexity lower bound for the set disjointness problem, and is similar to the arguments from Sanford et al. [2024c], Yehudai et al. [2024]. The lower bound for communication complexity is the following: Alice and Bob are given inputs $a, b \in \{0, 1\}^s$ respectively, and their goal is to find $\max a_i b_i$ by sending single bit-messages to each other in a sequence of communication rounds. The lower bound says that any deterministic protocol for solving such a task requires at least $s$ rounds of communication.

We set $s = n^2$, and design a graph $G = (V, E)$ that has a directed 2-cycle iff $\max a_i b_i = 1$. The graph has $|V| = 2n$, we partition the vertices into 2 disjoint sets $V_1, V_2$, and number the vertices of each set between 1 and $n$. The inputs $a$ and $b$ encode the adjacency matrices between vertices in $V_1$ and $V_2$, and between vertices in $V_2$ and $V_1$ respectively. Now, there exists a directed 2-cycle iff there is some $i \in [s]$ for which both $a_i = 1$ and $b_i = 1$ meaning that $\max a_i b_i = 1$.

Assume there exists a transformer of depth $L$ with $H$ heads, embedding dimension $m$ and bit precision $p$ that successfully detects 2-cycles in a directed graph. Denote the weights of head $i$ in layer $\ell$ by $Q_i^\ell, K_i^\ell, V_i^\ell \in \mathbb{R}^{m \times m}$ for each $i \in [H]$, and assume w.l.o.g. that they are of full rank, otherwise our lower bound would include the rank of these matrices instead of the embedding dimension (which can only strengthen the lower bound). We design a communication protocol for Alice and Bob to solve the set disjointness problem. The communication protocol will depend on whether $T$ has residual connections or not. We begin with the case that it does have them, the protocol works as follows:

1. Given input sequences $a, b \in \{0, 1\}^s$ to Alice and Bob respectively, they calculate the input tokens $x_1^0, \dots, x_n^0$ and $x_{n+1}^0, \dots, x_{2n}^0$, respectively. Note that the adjacency matrix have a block shape, thus both Alice and Bob can calculate the rows of the adjacency matrix corresponding the the edges which are known to them.

2. Bob calculates $K_j^1 x_i^0, Q_j^1 x_i^0, V_j^1 x_i^0$ for every head $j \in [H]$ and transmits them to Alice. The number of transmitted bits is $O(nmHp)$

3. Alice can now calculate the output of the $r$-th token after the first layer. Namely, for every head $j \in [H]$, she calculates:

$$s_j^r = \sum_{i=1}^{2n} \exp(x_i^{0\top} K_j^{1\top} Q_j^1 x_r^0)$$

$$t_j^r = \sum_{i=1}^{2n} \exp(x_i^{0\top} K_j^{1\top} Q_j^1 x_r^0) V_j^1 x_i^0 \ .$$

The output of the $j$-th head on the $r$-th token is equal to $\frac{t_j^r}{s_j^r}$. For the first $n$ tokens, Alice use the residual connection which adds the tokens that are known only to her. She now passes the tokens through the MLP to calculate $x_1^1, \ldots, x_n^1$, namely the output of the tokens known to her after the first layer.

4. Similarly to the previous 2 steps, Bob calculates the tokens $x_{n+1}^1, \ldots, x_{2n}^1$ which are known only to him.

5. For any additional layer, the same calculations are done so that Alice calculates $x_1^\ell, \ldots, x_n^\ell$ and Bob calculates $x_{n+1}^\ell, \ldots, x_{2n}^\ell$.

In case there are no residual connections, after the third step above Alice have the information about all the tokens. Hence, there is no need for more communication rounds, and Alice can finish the rest of the calculations of the transformers using the output tokens of the first layer.

By the equivalence between the set disjointness and the directed 2-cycle that was described above, Alice returns 1 iff the inputs $\max_i a_i b_i = 1$, and 0 otherwise. The total number of bits transmitted in this protocol in the case there are residual connection is $O(nmpHL)$, since there are $O(nmpH)$ bits transferred in each layer. The lower bound is determined by the size of the input which is $s = n^2$, hence $mpHL = \Omega(n)$. In the case there are no residual connections there is no dependence on $L$, hence the lower bound becomes $mpH = \Omega(n)$. $\qquad\square$

### B.3.1 An extension to bounded degree graphs

In order to prove the optimality result in Theorem 4.4 for the task of determining the existence of a 2-cycle in **bounded-degree graphs**, we state the following theorem.

**Theorem B.3.** *Let $T$ be a transformer with embedding dimension $m$ depth $L$, bit-precision $p$ and $H$ attention heads in each layer. If $T$ can detect $2$-cycles on $d$-degree directed graphs, then:*

1. *If $T$ has residual connections then $mpHL = \Omega(d)$.*

2. *If $T$ doesn't have residual connections then $mpH = \Omega(d)$.*

*Proof.* The proof is nearly identical to that of Theorem 4.2, except that we alter the reduction to ensure that that the graph possessed by Alice and Bob is of degree $d$.

As before, we design a graph $G = (V, E)$ with vertices partitioned into two sets $V_1, V_2$ satisfying $|V_1| = |V_2| = n$. Let $\bar{E}_d$ denote the edges of a bipartite graph between $V_1$ and $V_2$ such that (1) every node has $d$ incident outgoing edges; and (2) $(i, j) \in \bar{E}_d$ if and only if $(j, i) \in \bar{E}_d$.

Consider some instance of set disjointness with $a, b \in \{0, 1\}^s$ for $s = nd$. We index the $2s$ edges in $\bar{E}_d$ as $e_1^a = (v_1^1, v_1^2), \ldots, e_s^b = (v_s^1, v_s^2)$ and $e_1^b = (v_1^2, v_1^1), \ldots, e_s^b = (v_s^2, v_s^2)$. Then, we embed the instance by letting $e_i^a \in E$ if $a_i = 1$ and $e_i^b \in E$ if $b_i = 1$. As before, there exists a directed 2-cycle in $G$ if and only if $\max_i a_i b_i = 1$.

The analysis of the transformer remains unchanged. An $L$-layer transformer with embedding dimension $m$, heads $H$, and bit precision $p$ transmits $O(nmpHL)$ bits between Alice and Bob. The hardness of set disjointness requires that at least $s = nd$ bits be transmitted, which means that it must be the case that $mphL = \Omega(d)$. $\qquad\square$

### B.4 Proof of Theorem 4.3

We will need the following lemma for the proof.

**Lemma B.4.** *Let $a_1, \ldots, a_k, b_1, \ldots, b_k \in \mathbb{R}$ where the $a_i$'s are distinct. There exists a 2-layer fully-connected neural network $N : \mathbb{R} \to \mathbb{R}$ with width $O(k)$ such that $N(a_i) = b_i$.*

*Proof.* Let $\delta = \min_i \in [k] |a_i - a_j|$, by the assumption $\delta > 0$. Let:

$$f_i(x) = \frac{1}{\delta} \left( \sigma(x - (a_i - 2\delta)) - \sigma(x - (a_i - \delta) + \sigma(a_i + 2\delta - x) - \sigma(a_i + \delta - x) \right) .$$

It is clear that $f_i(a_i) = 1$ and $f_i(a_j) = 0$ for any $j \neq i$. Thus, we define the network: $N(x) = \sum_{i=1}^{k} b_i f_i(x)$. □

We are now ready to prove the theorem.

**Theorem 4.3.** *There exists an $O(L)$-layer transformer with embedding dimension $m = O(n)$ such that, for any graph embedded as rows of an adjacency matrix A, the output of the transformer in the $i$-th token is the $i$-th row of $A^L$.*

*Proof.* We will first show the construction for the case of $L = 1$, and then show inductively for general $L$. We define the input for the transformer as $X = \begin{pmatrix} A \\ I \\ \mathbf{0}_{d \times d} \\ \mathbf{0}_{d \times d} \end{pmatrix} \in \mathbb{R}^{3d \times d}$, namely, there are $d$ tokens, each token contains a column of the adjacency matrix concatenated with a positional embedding. The self-Attention layer contains one head with the following matrices:

$$K = c \cdot \begin{pmatrix} I & & \\ & \mathbf{0}_{d \times d} & \\ & & \mathbf{0}_{d \times d} \end{pmatrix}, \quad Q = \begin{pmatrix} \mathbf{0}_{d \times d} & I & \\ \mathbf{0}_{d \times d} & \mathbf{0}_{d \times d} & \\ & & \mathbf{0}_{d \times d} \end{pmatrix}, \quad V = \begin{pmatrix} I & & \\ & I & \\ & & \mathbf{0}_{d \times d} \end{pmatrix}.$$

where $c > 0$ is some sufficiently large constant the determines the temperature of the softmax. We first have that $X^\top K^\top Q X = A$. Since all the values of $A$ are either $0$ or $1$, for a sufficiently large $c > 0$, the softmax behave similarly to the hardmax and we get that:

$$V X \text{SM}(A) = \begin{pmatrix} A^2 \cdot \deg(A)^{-1} \\ A \cdot \deg(A)^{-1} \\ \mathbf{0}_{d \times d} \end{pmatrix},$$

where $\deg(A)$ is a diagonal matrix, where its $i$-th diagonal entry is equal to the degree of node $i$. Finally, we apply an MLP $\mathcal{N} : \mathbb{R}^{3d \times d} \to \mathbb{R}^{3d \times d}$ that operates on each token separately. We define the MLP such that:

$$\mathcal{N}\left( \begin{pmatrix} A^2 \cdot \deg(A)^{-1} \\ A \cdot \deg(A)^{-1} \\ \mathbf{0}_{d \times d} \end{pmatrix} \right) = \begin{pmatrix} \mathbf{0}_{d \times d} \\ \mathbf{0}_{d \times d} \\ A^2 \end{pmatrix}.$$

Constructing such an MLP can be done by calculating the degree of each token from $A \cdot \deg(A)^{-1}$ and multiplying the first $d$ coordinates of each token by this degree. This can be done since the entries of this matrix is either $0$ or the inverse of the degree of node $i$, thus it requires only inverting an integer between $1$ and $n$. By Lemma B.4 this can be done by a 2-layer MLP with width $n$. This finishes the construction for calculating $A^2$.

For general $L > 2$ we use the residual connection from the inputs. That is, the input to the $L$-th layer of the transformer is equal to $\begin{pmatrix} A \\ I \\ A^L \end{pmatrix}$. We use a similar construction as the above, except that we use the matrix $V = \begin{pmatrix} \mathbf{0}_{d \times d} & & \\ & I & \\ & & I \end{pmatrix}$. This way, the output of the self-attention layer is $\begin{pmatrix} \mathbf{0}_{d \times d} \\ A \cdot \deg(A)^{-1} \\ A^{L+1} \cdot \deg(A)^{-1} \end{pmatrix}$, and we employ a similar MLP as before to eliminate the $\deg(A)^{-1}$ term. □

## B.5 Proof of Theorem 4.4

**Theorem 4.4.** *For any $n \in \mathbb{N}$ and $d \leq n$, there exists a single-layer transformer with embedding dimension $O(d \log n)$ that detects 2-cycles in any graph with node degree at most $d$. This embedding dimension is optimal up to logarithmic factors.*

The optimality result is proved using the same methodology as Theorem 4.2. It is stated and proved formally as Theorem B.3.

The proof of the construction adapts an argument from Theorem 2 of Sanford et al. [2024c], which shows that a sparse averaging task can be solved with bounded-width transformers. We make use of the following fact, which is a consequence of the Restricted Isometry Property analysis of Candes and Tao [2005], Mendelson et al. [2005].

**Lemma B.5.** *For any $d \leq n \in \mathbb{N}$ and $p = \Omega(d \log n)$, there exist vectors $\mathbf{y}_1, \ldots, \mathbf{y}_n \in \mathbb{R}^p$ such that for any $\mathbf{x} \in \{0, 1\}^n$ with $\sum_i \mathbf{x}_i \leq d$, there exists $\phi(\mathbf{x}) \in \mathbb{R}^p$ such that*

$$\langle \phi(\mathbf{x}), \mathbf{y}_i \rangle = 1, \quad \textit{if } \mathbf{x}_i = 1,$$
$$\langle \phi(\mathbf{x}), \mathbf{y}_i \rangle \leq \frac{1}{2}, \quad \textit{if } \mathbf{x}_i = 0.$$

We use this fact to prove Theorem 4.4.

*Proof.* Concretely, we prove that some transformer $T$ exists that takes as input

$$X = \begin{pmatrix} \mathbf{x}_1 & \cdots & \mathbf{x}_n \\ 1 & \cdots & n \end{pmatrix} \in \mathbb{R}^{(n+1) \times n}$$

and returns $T(X) \in \{0, 1\}^n$, where $T(X_i) = 1$ if and only if the $i$th node in the graph whose adjacency matrix is $A$ belongs to a directed 2-cycle. We assume that no self-edges exist.

We first configure the input MLP to incorporate the above vectors for node identifiers and adjacency rows and produce the tokens $\tilde{X} = (\tilde{\mathbf{x}}_1, \ldots \tilde{\mathbf{x}}_n) \in \mathbb{R}^{m \times n}$ for $m = 2p + 2$:

$$\tilde{\mathbf{x}}_i = \begin{pmatrix} \phi(\mathbf{x}_i) \\ \mathbf{y}_i \\ 1 \\ 0 \end{pmatrix}.$$

We also introduce a constant-valued "dummy node" $\tilde{\mathbf{x}}_{n+1}$, which has no edges and does not appear in the output:

$$\tilde{\mathbf{x}}_{n+1} = \begin{pmatrix} \mathbf{0}_p \\ \mathbf{0}_p \\ 0 \\ 1 \end{pmatrix}.$$

We define linear transforms $Q, K, V \in \mathbb{R}^{d \times n}$ that satisfy the following, for any $i \in [n]$ and some sufficiently large temperature constant $c$:

$$Q\tilde{\mathbf{x}}_i = c \begin{pmatrix} \phi(\mathbf{x}_i) \\ \mathbf{y}_i \\ \frac{7}{4} \\ 0 \end{pmatrix}, \quad K\tilde{\mathbf{x}}_i = \begin{pmatrix} \mathbf{y}_i \\ \phi(\mathbf{x}_i) \\ 0 \\ 0 \end{pmatrix}, \quad K\tilde{\mathbf{x}}_{n+1} = \begin{pmatrix} \mathbf{0}_p \\ \mathbf{0}_p \\ 1 \\ 0 \end{pmatrix}, \quad V\tilde{\mathbf{x}}_i = \begin{pmatrix} \mathbf{0}_p \\ \mathbf{0}_p \\ 0 \\ 1 \end{pmatrix}, \quad V\tilde{\mathbf{x}}_{n+1} = \mathbf{0}_m,$$

Then, for any $i, j \in [n]$ with $i \neq j$, the individual elements of the query-key product are exactly

$$(\tilde{X}^\top K^\top Q \tilde{X})_{j,i} = c \left( \langle \phi(\mathbf{x}_i), \mathbf{y}_j \rangle + \langle \phi(\mathbf{x}_j).\mathbf{y}_i \rangle \right).$$

By applying Lemma B.5, we find that

$$(\tilde{X}^\top K^\top Q \tilde{X})_{j,i} = 2c, \quad \text{if } \mathbf{x}_{i,j} = 1 \text{ and } \mathbf{x}_{j,i} = 1;$$
$$(\tilde{X}^\top K^\top Q \tilde{X})_{j,i} \leq \frac{3}{2}c, \quad \text{otherwise.}$$

In contrast, $(\tilde{X}^\top K^\top Q \tilde{X})_{n+1,i} = \frac{7}{4}c$ for any $i$.

Thus, for sufficiently large $c$, all nonzero elements (after rounding) of $\text{SM}(\tilde{X}^\top K^\top Q \tilde{X})_{.,i}$ belongs to indices $j \in [n]$ if there exist at least one 2-cycle containing node $i$; if not, then $\text{SM}(\tilde{X}^\top K^\top Q \tilde{X})_{n+1,i} = 1$ and all others are zero.

By our choice of value vectors, the $i$th output of the self-attention unit is $\mathbf{e}_m$ if there exists a 2-cycle and $\mathbf{0}_m$ otherwise. $\qquad \square$

# C Proofs from Section 5

## C.1 Proof of Theorem 5.2

**Theorem 5.2.** *Let $k, n \in \mathbb{N}$, and let $G'$ be a graph with $k$ nodes. There exists a transformer with $O(1)$ self-attention layers and embedding dimension $O\left(n^{2-1/k}\right)$ that, for any graph $G$ of size $n$, counts the number of occurrences of $G'$ as a subgraph of $G$.*

*Proof.* The main bulk of the proof will use the transformer to prepare the inputs. We will first explain the layout of the construction, and then present it formally. Each input node is represented as a row of the adjacency matrix. We will split the nodes into $n^{1/k}$ sets, where each set contains $n^{1-1/k}$ nodes. The first layer will prepare the adjacency rows so that each token will include only edges of other nodes from the same set. The second layer will combine all the nodes of each set into a separate token. This will use $n^{1/k}$ tokens, where each of them will contain at most $n^{2-2/k}$ edges. The last layer will use each token to represent each possible combination of $k$ such sets. There are at most $\binom{n^{1/k}}{k} \leq n$ such combinations, and each of them contains at most $n^{2-2/k}$ edges. We need an additional $n^{1/k}$ entries for technical reasons to do this embedding into all possible combinations of sets.

We now turn to the formal construction. Assume that the nodes are numbered as $v_1, \ldots, v_n$, and denote by $\mathbf{x}_1, \ldots, \mathbf{x}_n$ the row of the adjacency matrix corresponding to the nodes. The input to the transformer of the node $v_i$ will be $\begin{pmatrix} \mathbf{x}_i \\ i \end{pmatrix} \in \mathbb{R}^{n+1}$, where $\mathbf{e}_i$ is the $i$-th standard unit vector.

Throughout the proof we assume that $n^{1/k}$ and $n^{1-1/k}$ are integers. Otherwise, replace them by their integral value.

**Layer 1:** We begin the construction with an MLP that operates on each token separately. This can be viewed as if we use the self-attention layer to have no effect on the inputs, by setting $V = 0$ and using the residual connection. The MLP will implement the following function:

$$\mathbb{R}^{n+1} \ni \begin{pmatrix} \mathbf{x}_i \\ i \end{pmatrix} \mapsto \begin{pmatrix} \tilde{\mathbf{x}}_i \\ \mathbf{w}_i \\ \mathbf{z}_i \\ i \end{pmatrix} \in \mathbb{R}^{n^{2-2/k}+2n^{1/k}+1} \ .$$

Intuitively, we split the nodes into $n^{1/k}$ sets, each one containing $n^{1-1/k}$ nodes. $\tilde{\mathbf{x}}_i$ will include a pruned adjacency row for node $i$ with only edges from its own set. $\mathbf{w}_i$ indicates to which set each node belongs to, and $\mathbf{z}_i$ indicate on the tokens that will store these sets. We first introduce the vectors $\bar{\mathbf{x}}_i \in \mathbb{R}^{n^{1-1/k}}$ that are equal to:

$$(\bar{\mathbf{x}}_i)_j = \sum_{r=1}^{n^{1/k}} \mathbb{1}((\mathbf{x}_i)_{(r-1)n^{1-1/k}+j} = 1) \cdot \mathbb{1}((r-1)n^{1-1/k}+1 \leq i \leq rn^{1-1/k}) \cdot \mathbb{1}((r-1)n^{1-1/k}+1 \leq j \leq rn^{1-1/k})$$

These vectors can be constructed using a 3-layer MLP. First, note that this function in our case operates only on integer value inputs, since $i, j$ and all the entries of $\mathbf{x}_i$ are integers, hence it is enough to approximate the indicator function up to a uniform error of $\frac{1}{2}$ and it will suffice for our purposes. To this end, we define the function:

$$f_{r,s}(z) = \sigma(x - (r-1)) - \sigma(x - r) + \sigma(x - (s+1)) - \sigma(x - s) \ .$$

Here $\sigma = \max\{0, z\}$ is the ReLU function. If $s \geq r + 2$ We get that $f_{r,s}(z) = 1$ for $r \leq z \leq s$, and $f_{r,s}(z) = 0$ for $z \leq r - 1$ or $z \geq s + 1$. This shows that the functions inside the indicators can be expressed (for integer valued inputs) using a 2-layer MLP. Expressing the multiplication of the indicators can be done using another layer:

$$g(z_1, z_2, z_3) = \sigma(z_1 + z_2 + z_3 - 2) = \mathbb{1}(z_1 = 1) \cdot \mathbb{1}(z_2 = 1) \cdot \mathbb{1}(z_3 = 1) \ ,$$

where $z_1, z_2, z_3 \in \{0, 1\}$. The width of this construction is $O(n)$ since for each of the $n^{1-1/k}$ coordinates of the output we sum $n^{1/k}$ such functions as above.

We define $\tilde{\mathbf{x}}_i \in \mathbb{R}^{n^{2-2/k}}$ for $i \equiv j \pmod{n^{1-1/k}}$ to be equal to $\bar{\mathbf{x}}_i$ in the coordinates $(j-1)n^{1-1/k}+1$ until $jn^{1-1/k}$ and all the other coordinates are $0$. These vectors will later be summed together across

all nodes in the same set, which provides an encoding of all the edges in the set. We also define $\mathbf{w}_i = \mathbf{e}_j \in \mathbb{R}^{n^{1/k}}$ for $(j-1)n^{1/k} + 1 \leq i \leq jn^{1/k}$ and $\mathbf{z}_i = \mathbf{e}_i \in \mathbb{R}^{n^{1/k}}$ for $i = 1, \ldots, n^{1/k}$ and $\mathbf{z}_i = \mathbf{0}$ otherwise.

**Layer 2:** We define the weights of the second layer of self-attention in the following way:

$$K = \begin{pmatrix} \mathbf{0}_{n^{2-2/k} \times n^{2-2/k}} & & & \\ & \mathbf{0}_{n^{1/k} \times n^{1/k}} & & \\ & & I_{n^{1/k}} & \\ & & & 0 \end{pmatrix},$$

$$Q = \begin{pmatrix} \mathbf{0}_{n^{2-2/k} \times n^{2-2/k}} & & & \\ & I_{n^{1/k}} & & \\ & & \mathbf{0}_{n^{1/k} \times n^{1/k}} & \\ & & & 0 \end{pmatrix},$$

$$V = \begin{pmatrix} n^{1/k} I_{n^{2-2/k}} & \\ & \mathbf{0}_{(2n^{1/k}+1) \times (2n^{1/k}+1)} \end{pmatrix}.$$

Given two vectors $\begin{pmatrix} \tilde{\mathbf{x}}_i \\ \mathbf{w}_i \\ \mathbf{z}_i \\ i \end{pmatrix}, \begin{pmatrix} \tilde{\mathbf{x}}_j \\ \mathbf{w}_j \\ \mathbf{z}_j \\ j \end{pmatrix}$, which are outputs of the previous layer, we have that:

$$\begin{pmatrix} \tilde{\mathbf{x}}_i \\ \mathbf{w}_i \\ \mathbf{z}_i \\ i \end{pmatrix}^{\top} K^{\top} Q \begin{pmatrix} \tilde{\mathbf{x}}_j \\ \mathbf{w}_j \\ \mathbf{z}_j \\ j \end{pmatrix} = \langle \mathbf{z}_i, \mathbf{w}_j \rangle.$$ This shows that the first $n^{1/k}$ tokens, which represent each of the $n^{1/k}$ sets, will attend with a similar weight to every node in their set. After applying the $V$ matrix we use the residual connection only for the positional embedding vectors[7] (namely, the last $2n^{1/k} + 1$ coordinates). Thus, the output of the self-attention layer for the first $n^{1/k}$ tokens encodes all the edges in their set in their first $n^{2-2/k}$ coordinates. This encoding is such that there is 1 in the $i$-th coordinates if there is an edge between nodes $v_s$ and $v_r$ in the set for where $r$ and $s$ are the unique integers such that $i \equiv r \pmod{n^{1-1/k}}$ and $(s-1)n^{2-2/k} + 1 < i < sn^{2-2/k}$. Thus, the output of the self-attention layer can be written as $\begin{pmatrix} \mathbf{y}_i \\ \mathbf{w}_i \\ \mathbf{z}_i \\ i \end{pmatrix} \in \mathbb{R}^{n^{2-1/k}+2n^{1/k}+1}$, where $\mathbf{y}_i$ is either an encoding as described above (for $i \leq n^{1/k}$) or some other vector (for $i \geq n^{1/k}$) for which its exact value will not matter. The vector $\mathbf{w}_i$ is a positional embedding that is not needed anymore and will be removed by the MLP, and $\mathbf{z}_i = \mathbf{e}_i$ for $i \leq n^{1/k}$ and $\mathbf{z}_i = \mathbf{0}$ otherwise.

We will now construct the MLP of the second layer. First, note that given $n^{1/k}$ sets, the number of all $k$ combinations of such sets is bounded by $\binom{n^{1/k}}{k} \leq n^{k \cdot 1/k} = n$. Denote all possible combinations by $B_1, \ldots, B_n$ and let $\mathbf{v}_1, \ldots, \mathbf{v}_n \in \mathbb{R}^{n^{1/k}}$ such that $(\mathbf{v}_i)_j = 1$ if $B_i$ includes the $j$-th set, and 0 otherwise. These vectors encode all the possible combinations of such sets. The MLP will apply the following map:

$$\mathbb{R}^{n^{2-1/k}+2n^{1/k}+1} \ni \begin{pmatrix} \mathbf{y}_i \\ \mathbf{w}_i \\ \mathbf{z}_i \\ i \end{pmatrix} \mapsto \begin{pmatrix} \mathbf{y}_i \\ \mathbf{v}_i \\ \mathbf{z}_i \end{pmatrix} \in \mathbb{R}^{n^{2-1/k}+2n^{1/k}+1}.$$

This map can be implemented by a 3-layer MLP. Specifically, the only part coordinates that changes are those of $\mathbf{w}_i$ which are replaced by $\mathbf{v}_i$. This can be done using the function $f(i) = \sum_{j=1}^{n} \mathbb{1}(i = j) \cdot \mathbf{v}_j$, and its construction is similar to the construction of the MLP in the previous layer.

---

[7] It is always possible to use the residual connection to affect only a subset of the coordinates. This can be done by doubling the number of unaffected coordinates, using the $V$ matrix to move the unaffected entries to these new coordinates, and then using a 1-layer MLP to move the unaffected entries to their previous coordinates (which now include what was added through the residual connection). We omit this construction from here for brevity and since it only changes the embedding dimension by a constant factor.

**Layer 3:** The last self-attention layer will include the following weight matrices:

$$K = \begin{pmatrix} \mathbf{0}_{n^{2-1/k} \times n^{2-1/k}} & & \\ & I_{n^{1/k}} & \\ & & \mathbf{0}_{n^{1/k} \times n^{1/k}} \end{pmatrix},$$

$$Q = \begin{pmatrix} \mathbf{0}_{n^{2-1/k} \times n^{2-1/k}} & & \\ & \mathbf{0}_{n^{1/k} \times n^{1/k}} & \\ & & I_{n^{1/k}} \end{pmatrix},$$

$$V = \begin{pmatrix} kI_{n^{2-1/k}} & \\ & \mathbf{0}_{(2n^{1/k}+1) \times (2n^{1/k})} \end{pmatrix}.$$

After applying this layer to the outputs of the previous layer, each token $i$ will attend, with similar weight, to all the sets (out of the $n^{1/k}$ sets of nodes) that appear in its positional embedding vector $\mathbf{v}_i$. Thus, after applying this layer, The first $n^{2-1/k}$ contain an encoding (as described in the construction of the previous layer) of all the edges in the $i$-th combination of $k$ sets $B_i$.

Finally, the MLP will be used to detect whether the given subgraph of size $k$ appears as a subgraph in the input graph (which is an encoding of the edges). The output of the MLP will be 1 if the subgraph appears and 0 otherwise.

Note that any subgraph of size $k$ must appear in one of those combination of sets. Thus, by summing all the tokens, if their sum is greater than 0 the subgraph of size $k$ appears as a subgraph of $G$.

$\square$

### C.2 Proof of Theorem 5.1

We first define the *Eulerian cycle verification problem* on multi-graphs.

Consider some directed multi-graph $G = (V, E)$ for $V = \{v_1, \ldots, v_n\}$ and $E = \{e_1, \ldots, e_N\}$, where each edge is labeled as

$$e_j = (e_{j,1}, e_{j,2}, j) \in V \times V \times [N].$$

We say that $e_j$ is a *successor edge* of $e_i$ if $e_{i,2} = e_{j,1}$. A problem instance also contains a fragmented path, which is expressed as a collection of ordered pairs of edges $P = \{p^1, \ldots, p^N\}$ with

$$p^j = (p_1^j, p_2^j) \in E \times E,$$

where $p_1^j$ and $p_2^j$ are successive edges (i.e. $p_{1,2}^j = p_{2,1}^j$). Let $p^j$ be a *successor path fragment* of $p^i$ if $p_2^i = p_1^j$.

We say that $P$ *verifies an Eulerian cycle* if

1. every edge in $E$ appears in exactly two pairs in $P$; and

2. there exists a permutation over pairs $\sigma : [N] \to [N]$ such that each $p^{\sigma(j+1)}$ is a successor of $p^{\sigma(j)}$ (and $p^{\sigma(0)}$ is a successor of $p^{\sigma(N)}$).

We treat Eulerian cycle detection as a sequential task on adjacency-node tokenization inputs by setting the $i$th embedding to $\phi(v_i, P_i)$, where $P_i$ encodes all pairs incident to node $v_i$, i.e.,

$$P_i = \{p \in P : p_{1,2} = p_{2,1} = v_i\}.$$

Now, we prove that—conditional on the hardness of distinguishing one-cycle and two-cycle graphs— no transformer can solve the Eulerian cycle verification problem of degree-$n$ multi-graphs without sufficient width or depth.

**Theorem 5.1.** *Under Conjecture 2.4 from Sanford et al. [2024b], the Eulerian cycle verification problem on multigraphs with self loops cannot be solved by transformers with adjacency matrix inputs if $m = O\left(n^{2-\epsilon}\right)$ for any constant $\epsilon > 0$, unless $L = \Omega(\log(n))$.*

*Proof.* Consider some transformer $T$ with depth $L$ and embedding dimension $m$ that solves the Eulerian cycle verification problem for any *directed* multi-graph with $n$ nodes and at most $n^2$ edges.

We use this to construct a transformer $\mathfrak{T}$ with depth $L + O(1)$ and embedding dimension $O(m + n^{1.1})$ that distinguishes between an *undirected*[8] cycle graph of size $N$ and two cycles of size $\frac{N}{2}$ for $N = \frac{n^2}{2}$. The claim of the theorem follows as an immediate consequence of the 1 vs. 2 cycle conjecture (as stated in Conjecture 13 of Sanford et al. [2024a]).

We prove that a transformer with $O(1)$ layers and embedding dimension $O(n)$ can convert an $N$-node cycle graph instance $\mathfrak{G} = (\mathfrak{V}, \mathfrak{E})$ into a multi-graph $G = (V, E)$ with paths $P$ such that $\mathfrak{G}$ is a single cycle if and only if $P$ represents an Eulerian path on $G$. We first define the transformation and then show that it can be implemented by a small transformer.

- Assume without loss of generality that $\mathfrak{V} = [N]$ and $V = [n]$. Let $\phi_n(i) = i \pmod{\frac{n}{2}}$ be a many-to-one mapping from vertices in $\mathfrak{V}$ to half of the vertices in $V$.

- For each undirected edge $\mathfrak{e}_i = \{\mathfrak{v}_1, \mathfrak{v}_2\} \in \mathfrak{E}$, we add two directed edges to $E$:
$$e_i = (\phi_n(\mathfrak{v}_1), \phi_n(\mathfrak{v}_2), i), \text{ and } e_{-i} = (\phi_n(\mathfrak{v}_2), \phi_n(\mathfrak{v}_1), -i).$$
  For an arbitrary *turnaround edge* edge $\mathfrak{e}_i = \mathfrak{e}^* \in \mathfrak{E}$, we replace $\mathfrak{e}_{i^*}, \mathfrak{e}_{-i^*}$ with two self edges:
$$e_i = (\phi_n(\mathfrak{v}_2), \phi_n(\mathfrak{v}_2), i), \text{ and } e_{-i} = (\phi_n(\mathfrak{v}_1), \phi_n(\mathfrak{v}_1), -i).$$

- For edge $\mathfrak{e}_i \in \mathfrak{E}$ as above with unique neighbors $\mathfrak{e}_j = \{\mathfrak{v}_0, \mathfrak{v}_1\}$ and $\mathfrak{e}_k = \{\mathfrak{v}_2, \mathfrak{v}_3\}$, we add paths $p^i = (e_i, e_{ak})$ and $p^{-i} = (e_{-i}, e_{bj})$, where $a$ and $b$ are chosen such that $e_{ak}$ and $e_{bj}$ succeed $e_i$ and $e_{-i}$ respectively.

We remark on a few properties of the constructed graph $G$, which satisfies $|V| = n$, and $|E| = |P| = n^2$. Any two adjacent edges in $\mathfrak{G}$ create two "successor relationships" between pairs of path segments in $P$. Then, if a cyclic subgraph of $\mathfrak{G}$ *does not* contain $\mathfrak{e}^*$, the path segments in $P$ produced by the edges in the subgraph comprise two disjoint directed cycle paths. On the contrary, if the subgraph contains $\mathfrak{e}^*$, then its path segments comprise a single cycle path.

Therefore, if $\mathfrak{G}$ is contains a single cycle of length $N$, then $P$ verifies an Eulerian cycle in $G$ that includes all $n^2$ edges Otherwise, if $\mathfrak{G}$ has two cycles of length $2N$, then $P$ represents *three* cyclic paths, one of length $\frac{n^2}{2}$ and two of length $\frac{n^2}{4}$. Hence, there is a one-to-one correspondence between the 1 vs. 2 cycle detection problem on $\mathfrak{G}$ and the Eulerian cycle verification problem on $G$ and $P$.

We conclude by outlining the construction of transformer $\mathfrak{T}$ that solves the cycle distinction problem. This can be implemented using elementary constructions or the existing equivalence between transformers and MPC.

- $\mathfrak{T}$ takes as input a stream of edges, $\mathfrak{e}_1, \ldots \mathfrak{e}_N \in \mathfrak{E}$, in no particular order. These are expressed in the *edge tokenization*, which means that the $i$th input to the transformer is $(\mathfrak{e}_{i,1}, \mathfrak{e}_{i,2}, i)$. We arbitrarily denote $\mathfrak{e}_1 = \mathfrak{e}^*$ with its positional embedding.

- In the first attention layer, $\mathfrak{T}$ retrieves the two adjacent edges of each edge embedding.

  It does so with two attention heads. The first encodes $\mathfrak{e}_{i,1}$ as a query vector (which is selected to be nearly orthogonal to those of each of the other $N$ edge embeddings), $\mathfrak{e}_{i,1} + \mathfrak{e}_{i,2}$ as a key vector, and $(\mathfrak{e}_{i,1}, \mathfrak{e}_{i,2}, i)$ as the value vector. The second does the same with $\mathfrak{e}_{i,2}$. $O(\log n)$ embedding dimension suffices for this association.

  Then, the $i$th output of this layer is processed by an MLP that computes $p^i$ and $p^{-i}$.

- The second attention layer collects all path tokens incident to node $j \in [n]$ (i.e., every $p^i \in P_j$) in the $j$th embedding. This can be treated as a single communication operation in the Massively Parallel Computation (MPC) model of Karloff et al. [2010], where $N$ machines each send $O(\log n)$ bits of information to $n$ machines, where each machine receives at most $O(n \log n)$ bits. Due to Theorem 1 of Sanford et al. [2024a], this attention layer can complete the routing task with embedding dimension $m = O(n^{1.1})$.

- Now, the $j$th element computes $\phi(v_j, P_j)$ and passes the embedding as input to $T$.

---

[8]The 1 vs. 2 cycle conjecture applies only to undirected graphs.

- If $T$ verifies an Eulerian cycle, let $\mathfrak{T}$ output that $\mathfrak{G}$ is a single-cycle graph.

Therefore, the existence of a transformer $T$ with depth $o(\log n)$ or width $O(N^{2-\epsilon})$ that solves the Eulerian cycle verification problem contradicts the 1 vs. 2 cycle conjecture. This completes the proof. □

# D    Additional Experiments

Here, we provide additional results of the experiment described in Section 6.1. The results for the 4-Cycle and Triangle Count with 50 and 100 nodes, as well as the connectivity task with 50 nodes, are presented in Figures 4, 5, 6, 7, 8. These experiments present the same trend discussed in Section 6.1, where the training loss and test accuracy and loss are similar, while training and inference times are drastically better for shallow-wide networks.

In Figure 9 we present the critical width chart as described in Section 6.1, for the Triangle Count task.

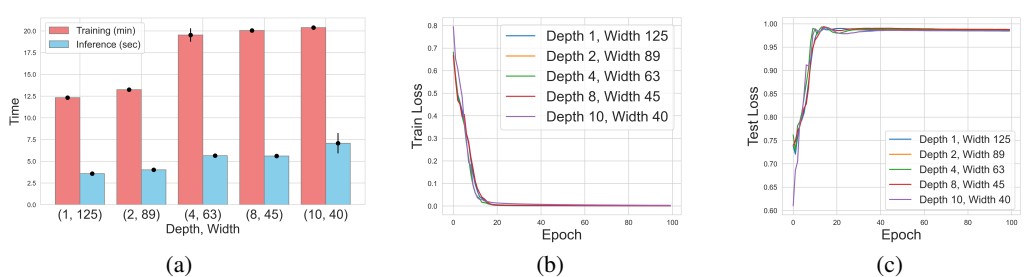

(a)                                    (b)                                    (c)

Figure 4: Training and inference times (a), training loss curves (b), and accuracy curves (c) for the 4-Cycle count task over graphs with 100 nodes, across transformers with approximately 100k parameters, varying in width and depth. While the loss and accuracies remain consistent, shallow and wide transformers demonstrate significantly faster training and inference times.

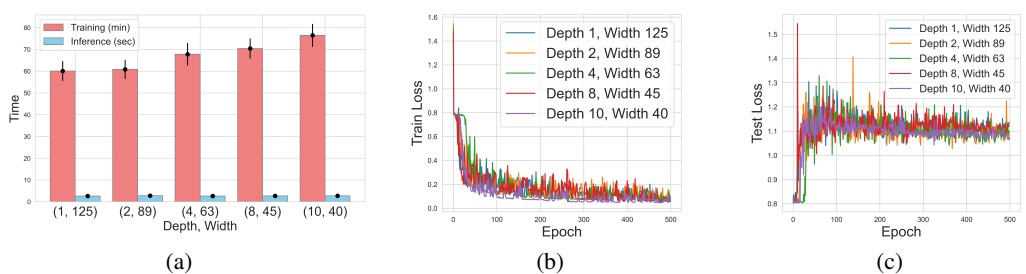

(a)                                    (b)                                    (c)

Figure 5: Training and inference times (a), training loss curves (b), and accuracy curves (c) for the 4-Cycle count task over graphs with 100 nodes, across transformers with approximately 100k parameters, varying in width and depth. While the loss and accuracies remain consistent, shallow and wide transformers demonstrate significantly faster training and inference times.

# E    Experimental Details

**Dataset information**    In Section 6.1 we used three synthetic datasets, including connectivity, Triangle Count and 4-Cycle Count. The Triangle Count and 4-Cycle Count were presented in Chen et al. [2020]. Each of these datasets contains 5000 graphs, and the number of nodes is set according to the configuration, as we tested graphs with increasing numbers of nodes. The counting datasets are generated using Erdős–Rényi graphs with an edge probability of 0.1.

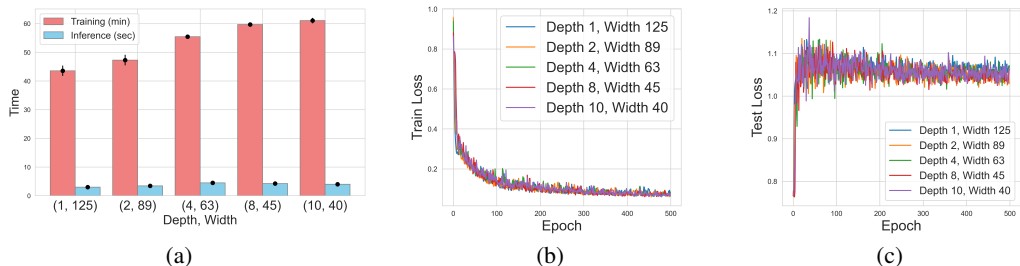

Figure 6: Training and inference times (a), training loss curves (b), and accuracy curves (c) for the 4-Cycle count task over graphs with 50 nodes, across transformers with approximately 100k parameters, varying in width and depth. While the loss and accuracies remain consistent, shallow and wide transformers demonstrate significantly faster training and inference times.

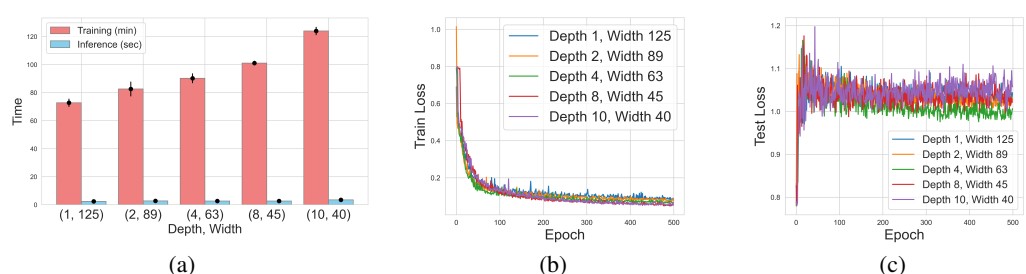

Figure 7: Training and inference times (a), training loss curves (b), and accuracy curves (c) for the Triangle Count task over graphs with 50 nodes, across transformers with approximately 100k parameters, varying in width and depth. While the loss and accuracies remain consistent, shallow and wide transformers demonstrate significantly faster training and inference times.

For the connectivity dataset, to avoid a correlation between connectivity and edge probability as exists in Erdős–Rényi graphs, we generated the datasets using diverse graph distributions, each with multiple distribution parameters. The dataset consists of graphs that are either connected or disconnected, generated using different random graph models to ensure diversity. The Erdős–Rényi model $G(n, p)$ is used, where each edge is included independently with probability $p$, and connected graphs are ensured by choosing $p \geq \frac{\ln n}{n}$, while disconnected graphs use a lower $p$. Random Geometric Graphs (RGGs) are also employed, where nodes are placed randomly in a unit space, and edges are formed if the Euclidean distance is below a certain threshold $r$; connected graphs use a sufficiently high $r$, whereas disconnected graphs are created with a lower $r$. Additionally, Scale-Free networks generated using the Barabási–Albert model are included, where new nodes attach preferentially to high-degree nodes, ensuring connectivity when enough edges per node ($m$) are allowed, while disconnected graphs are produced by limiting interconnections between components. Lastly, the Stochastic Block Model (SBM) is used to generate community-structured graphs, where intra-community connection probabilities ($p_{\text{intra}}$) are set high for connected graphs, and inter-community probabilities ($p_{\text{inter}}$) are set to zero to ensure disconnected graphs. Each type of graph is sampled in equal proportions, shuffled, and split into training, validation, and test sets to maintain class balance.

In Section 6.3 we used three molecular property prediction datasets from Open Graph Benchmark (OGB) Hu et al. [2020]. In ogbg-molhiv, the task is to predict whether a molecule inhibits HIV replication, a binary classification task based on molecular graphs with atom-level features and bond-level edge features. ogbg-bbbp involves predicting blood-brain barrier permeability, a crucial property for drug development, while ogbg-bace focuses on predicting the ability of a molecule to bind to the BACE1 enzyme, associated with Alzheimer's disease. Dataset statistics are presented in Table 2.

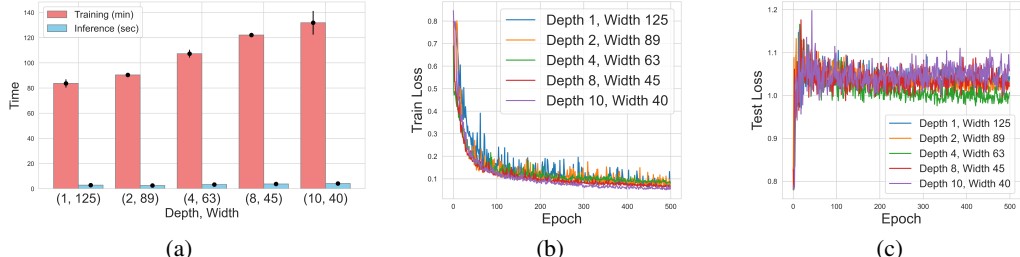

(a)           (b)           (c)

Figure 8: Training and inference times (a), training loss curves (b), and accuracy curves (c) for the Triangle Count task over graphs with 100 nodes, across transformers with approximately 100k parameters, varying in width and depth. While the loss and accuracies remain consistent, shallow and wide transformers demonstrate significantly faster training and inference times.

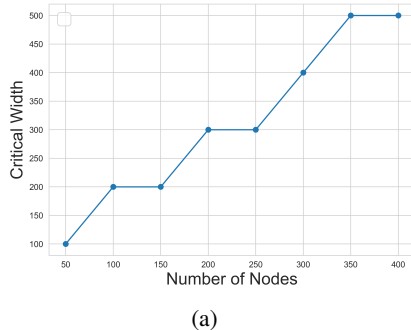

(a)

Figure 9: Critical width evaluation for the Triangle count Task. The points indicate the critical width at which the model fails to fit the data.

**Hyper-Parameters**  For all experiments, we use a fixed drouput rate of 0.1 and Relu activations. In Section 6.1 we tuned the learning rate in $\{10^{-4}, 5 \cdot 10^{-5}\}$, batch size in $\{32, 64\}$. In Section 6.3 we tuned the learning rate in $\{10^{-3}, 5 \cdot 10^{-3}\}$, number of layers in $\{3, 5, 6, 10, 12\}$, hidden dimensions in $\{32, 64\}$. We used batch size of size 64.

**Edge List tokenization**  Tokenization of the graph as a list of edges is done as follows. Assume a graph over $n$ nodes. Each node is represented by a one-hot encoding vector $b_i \in \mathbb{R}^n$ concatenated with the node input features $x_i \in \mathbb{R}^d$. Then, each edge is represented by concatenating its node representations. Each edge representation is fed as an independent token to the transformer. As graphs vary in size, we pad each node representation with zeros to match the maximal graph size in the dataset.

**Adjacency Rows tokenization**  Tokenization of the graph as an adjacency rows is done as follows. Assume a graph over $n$ nodes and adjacency matrix $A$. Each node is associated with a vector of features $x_i \in \mathbb{R}^d$. We concatenate to each row of $A$ the node's corresponding feature vector. This results in a vector of size $n + d$ for each node. As graphs vary in size, we pad each node vector with zeros to match the maximal graph size in the dataset. Each such vector is used as an input token to the transformer

**Laplacian Eigenvectors tokenization**  Tokenization of the graph as Laplacian eigenvectors is done as follows. Assume a graph over $n$ nodes and graph Laplacian $L$. Each node is associated with a vector of features $x_i \in \mathbb{R}^d$. We compute the eigenvector decomposition of the graph's Laplacian. Then, each node is associated an eigenvector and an eigenvalue. We concatenate these two for each node, resulting in a vector of size $n + 1$. We then concatenate to each of these spectral vectors the node's corresponding feature vector $x_i$. This results in a vector of size $n + d + 1$ for each node.

As graphs vary in size, we pad each node vector with zeros to match the maximal graph size in the dataset. Each such vector is used as an input token to the transformer

Table 2: Summary statistics of datasets used in Section 6.

| Dataset | # Graphs | Avg # Nodes | Avg # Edges | # Node Features | # Classes |
|---|---|---|---|---|---|
| ogbg-molhiv | 41,127 | 25.5 | 27.5 | 9 | 2 |
| ogbg-molbace | 1,513 | 34.1 | 36.9 | 9 | 2 |
| ogbg-molbbbp | 2,039 | 24.1 | 26.0 | 9 | 2 |

**Small and medium size graph commonly used datasets**   In the main paper we mentioned that many commonly used graph datasets contain graphs of relatively small size. Therefore in many real-world cases, the embedding dimension of the model is larger than the size of the graph. Here we list multiple such datasets from the Open Graph Benchmark (OGB) Hu et al. [2020] as well as TUdatasets Morris et al. [2020]. The datasets, including their statistics, are listed in Table 3.

Table 3: Summary of commonly used graph datasets, where the average number of nodes is relatively small

| Dataset | # Graphs | Avg # Nodes | Avg # Edges |
|---|---|---|---|
| ogbg-molhiv | 41,127 | 25.5 | 27.5 |
| ogbg-molbace | 1,513 | 34.1 | 36.9 |
| ogbg-molbbbp | 2,039 | 24.1 | 26.0 |
| ogbg-tox21 | 7,831 | 18.6 | 19.3 |
| ogbg-toxcast | 8,576 | 18.8 | 19.3 |
| ogbg-muv | 93,087 | 24.2 | 26.3 |
| ogbg-bace | 1,513 | 34.1 | 36.9 |
| ogbg-bbbp | 2,039 | 24.1 | 26.0 |
| ogbg-clintox | 1,477 | 26.2 | 27.9 |
| ogbg-sider | 1,427 | 33.6 | 35.4 |
| ogbg-esol | 1,128 | 13.3 | 13.7 |
| ogbg-freesolv | 642 | 8.7 | 8.4 |
| ogbg-lipo | 4,200 | 27.0 | 29.5 |
| IMDB-Binary | 1000 | 19 | 96 |
| IMDB-Multi | 1500 | 13 | 65 |
| Proteins | 1113 | 39.06 | 72.82 |
| NCI1 | 4110 | 29.87 | 32.3 |
| Enzymes | 600 | 32.63 | 62.14 |

