# OpenReview forum: "Depth-Width Tradeoffs for Transformers on Graph Tasks"
_NeurIPS.cc/2025/Conference — NeurIPS 2025 spotlight_

### Official Review · Reviewer_KP4t · 2025-06-30

**Clarity:** 2
**Significance:** 2
**Originality:** 2
**Rating:** 4
**Confidence:** 3

**Summary:**

This paper studies how transformer depth and width affect graph reasoning tasks. It shows that constant-depth, wide transformers can solve many tasks efficiently, with a theoretical hierarchy for width requirements. Using node-adjacency inputs, experiments confirm that shallow, wide models match deep ones in accuracy while training faster on both synthetic and real graph datasets.

**Questions:**

The graph inputs using adjacency rows resemble Graphormer-style encodings rather than token-based input sequences commonly used in transformers. The authors should clarify how these formats are aligned. The detailed questions are aligned with the weakness

Furthermore, they should explain why there is a significant performance gap between real-world applications, prior work, and the idealized results reported on synthetic tasks.

**Ethical Concerns:**

["NO or VERY MINOR ethics concerns only"]

**Final Justification:**

I still have concerns about whether converting the graphs into adjacency rows would affect the validity of the proofs. Nevertheless, I will maintain my positive score, as the main contribution is interesting.

**Limitations:**

yes

**Quality:**

3

**Strengths And Weaknesses:**

This paper presents a novel theoretical investigation into how transformer architectures—specifically their depth and width—affect their ability to solve graph algorithmic tasks. The key insight is that increasing the model's width can allow constant-depth transformers to solve a variety of graph problems, including tasks that previously required deeper architectures. The authors establish a theoretical hierarchy of width requirements and validate their findings with empirical evidence.

The work offers a valuable contribution toward understanding the algorithmic capabilities of transformers for graph tasks. However, several key aspects of the formulation and analysis remain unclear or underexplored, particularly in how graph representations are structured and interpreted by the transformer.

What exactly is the output for the 2-cycle detection task? Is it a binary classification task, or does it require identifying the specific edge pair forming the 2-cycle? This should be explicitly clarified in the experimental section.

The paper adopts a node-adjacency input format (each token is a row of the adjacency matrix), deviating from the sequential token-based representations used in prior works[1]. This raises the question: If node ordering changes, does the transformer still produce consistent predictions? If the answer is no, it calls into question the validity of experiments and the stability of results under permutation.

The paper states that each node's adjacency row is embedded as an input token. Does this imply that the transformer can read and interpret individual node features? A clearer explanation of how these embeddings are processed in the early layers is needed.

The claimed generalization across different input formats (e.g., edge list vs. adjacency row) seems questionable. In edge-list inputs, the model needs to merge multiple tokens corresponding to the same node, whereas in the adjacency-row setup, each node is represented once. How does the proposed theoretical framework reconcile these structural differences?

The molecular graph classification results reported are significantly below the performance of standard GNNs such as GCN or GAT (which often exceed 80% ACC)[2]. Given the synthetic tasks achieve near-perfect accuracy, the performance gap on real data deserves deeper analysis. Is the drop due to width, depth, tokenization method, or other modeling limitations?

In Section 4, please clarify whether the “n-node” and “n-edge”

[1]Sanford C, Fatemi B, Hall E, et al. Understanding transformer reasoning capabilities via graph algorithms[J]. Advances in Neural Information Processing Systems, 2024, 37: 78320-78370.

[2]Hu W, Fey M, Zitnik M, et al. Open graph benchmark: Datasets for machine learning on graphs[J]. Advances in neural information processing systems, 2020, 33: 22118-22133.

---

> ### Author Rebuttal · Authors · 2025-07-31
>
> We thank the reviewer for the thorough and positive review.
>
> Weaknesses:
>
> *“What exactly is the output for the 2-cycle detection task?”*: This is a binary classification task; the label is whether the input is a 1-cycle or 2-cycles. This is similar to the binary connectivity task, where the output is whether the graph is connected (1-cycle) or not (2-cycles).
>
> *“The paper adopts a node-adjacency input format “*: The node adjacency embeddings are not permutation invariant. It is common in theoretical papers about the expressiveness of transformers on graph tasks to use non-permutation invariant encoding to analyze their expressive power. This is also done, for example, in [1] and [2], where the authors use edge-list encoding, which is also not permutation invariant. We also emphasize that using any permutation invariant encoding, solving connectivity problems on graphs with $n$ nodes requires at least $\Omega(n)$ layers, since such models are equivalent to the LOCAL distributed computing architecture, see [3]. To utilize the expressive power of transformers, which is equivalent to the MPC model and allows solving connectivity problems with $O(\log(n))$ layers, we must use non-permutation invariant encoding. We will emphasize this point in the final version.
>
> *“The paper states that each node's adjacency row is embedded as an input token.”*: Yes, in the case of having node features (i.e., beyond combinatorial problems), each token also contains its node feature, and the transformer can interpret them in the early layers. We will clarify this in the final version.
> “The claimed generalization across different input formats”: Theoretically, it is possible to move from edge-list encoding to adjacency rows embedding using a constant-depth transformer. This transformer only needs to encode the edges relevant to each node and order them as an adjacency row. A translation in the other direction is also possible. We are happy to go into more details on this if the reviewer is interested. The conclusion of this is that for constant-depth transformers, theoretically, it doesn’t matter if we begin with an edge-list of adjacency-rows encoding, and our constructions will still work.
>
> *“The molecular graph classification results reported are significantly below the performance of standard GNNs”*: We agree with the reviewer on this point. The goal of this experiment is to justify the use of adjacency row encoding, compared to edge-list or spectral encodings. We believe that the drop in accuracy is since GNNs are currently more successful than transformers on these tasks in general. We also haven’t used the most powerful or largest transformer architecture that may achieve SOTA results on these tasks. We will consider adding experiments on GNNs to have a more thorough comparison.
>
>
> Questions:
>
> *“The graph inputs using adjacency rows resemble Graphormer-style encodings”*: We consider the rows of the adjacency metric as sequence tokens as used in Transformers, each corresponding to a node in the graph. Also in Graphormer, tokens of nodes (with some PEs) are eventually fed into a transformer as tokens, and therefore this can be seen in the same way.
>
> *“Furthermore, they should explain why there is a significant performance gap between real-world applications, prior work, and the idealized results reported on synthetic tasks.“*: Our experiments are not aimed at beating SOTA results, but rather to demonstrate and justify the use of the positional embedding we used. Thus, our hyperparameter grid and model size are smaller than those of SOTA models, which may be the cause of the difference. We will consider in the final version extending these experiments to larger models and attempting to achieve comparable results to the current SOTA.
>
>
> [1] C. Sanford, D. Hsu, and M. Telgarsky. Transformers, parallel computation, and logarithmic depth, 2024
>
> [2] C. Sanford, D. J. Hsu, and M. Telgarsky. Representational strengths and limitations of transformers, 2024
>
> [3] Loukas, Andreas. What graph neural networks cannot learn: depth vs width, 2019

---

> > ### Comment · Reviewer_KP4t · 2025-08-05
> >
> > Thank you for the authors’ detailed response. However, I still have concerns about whether converting the graphs into adjacency rows would affect the validity of the proofs. Nevertheless, I will maintain my positive score, as the main contribution is interesting.

---

### Official Review · Reviewer_hADS · 2025-07-02

**Clarity:** 3
**Significance:** 3
**Originality:** 3
**Rating:** 5
**Confidence:** 3

**Summary:**

The paper investigates the minimal size of a transformer needed to solve algorithmic problems, focusing specifically on graph-based tasks. It analyzes the expressive power of transformers under different depth and width settings. The authors show that with linear width and constant depth, transformers can handle tasks like connectivity and cycle detection, while more complex tasks require quadratic width. The paper offers both theoretical constructions and empirical evaluations to demonstrate these trade-offs across various graph encoding schemes.

**Questions:**

1. How do you see your theoretical results transferring to practical graph learning tasks beyond counting substructures (e.g., property prediction on molecular graphs)?

2. Have you considered adding comparisons with GNN baselines to better contextualize the empirical advantages of your transformer encodings?

3. Could you clarify how robust your theoretical constructions are to architectural and optimization constraints in real transformers?

**Ethical Concerns:**

["NO or VERY MINOR ethics concerns only"]

**Final Justification:**

Good paper.

**Limitations:**

Yes

**Quality:**

3

**Strengths And Weaknesses:**

Strengths:
1. This paper introduces a novel representational hierarchy that characterizes the relationship between depth and width in transformers for graph algorithmic tasks. The authors show that several problems—including connectivity and cycle detection—can be solved with fixed (constant) depth and linear width relative to the number of nodes.

2. The author proves that more complex tasks, such as subgraph counting and Eulerian cycle verification, require quadratic width. The work establishes both upper and lower bounds on the necessary width for a variety of graph problems, providing a hierarchy of task complexity.

3. The paper includes empirical results that demonstrate the practical advantages of using shallow, wide transformers in terms of training and inference time, while exploring different graph encoding schemes like adjacency rows, edge lists, and Laplacian eigenvectors.

Weaknesses:
1. The theoretical results and conclusions primarily focus on tasks like connectivity and triangle counting. While important for understanding algorithmic complexity, these tasks are relatively simple and may not generalize well to practical graph learning tasks such as graph classification, molecular property prediction, or social network analysis. This limits the direct applicability of the proposed width-depth trade-offs in real-world scenarios.

2. In the experimental section evaluating graph encodings, the authors do not include comparisons with state-of-the-art GNN architectures such as GCN, GIN, or GraphSAGE. Without these baselines, it's difficult to assess the practical competitiveness of the proposed transformer-based encodings and to understand the empirical relevance of the theoretical findings for practitioners choosing between GNNs and transformers.

3. The theoretical analysis assumes highly expressive MLP layers and precise control over attention weights to simulate operations such as computing matrix powers. While this is standard in theory papers, these assumptions may not hold in practice for real transformer architectures with finite width MLPs, limited numerical precision, and non-convex optimization. This gap between theory and practice limits the direct relevance of the derived width-depth bounds for deployed models.

---

> ### Author Rebuttal · Authors · 2025-07-31
>
> We thank the reviewer for the thorough and positive review.
>
> Weaknesses:
>
> 1) We addmitingly study theoretically only combinatorial problems on graphs. The advantage of such problems is that analyzing them is tractable, and also well connected to other fields in CS such as distributed computing, and graph algorithms. The difficulty of analyzing problems on real datasets, such as molecules and social networks, is that the properties of the graphs and the tasks themselves are rather difficult to formulate and analyze. It would be interesting to extract properties of such tasks and analyze them theoretically, but this is beyond the scope of our paper.
>
> 2) We indeed don’t compare our results to GNN architectures, and focus only on transformers. The goal of our paper is more of a theoretical analysis, and the experiment on the molecular dataset is to justify the use of adjacency encoding. With that said, we will consider adding more comparisons to GNN architectures in the final version.
>
> 3) The use of a highly expressive MLP and exact control of the attention weights was also done in previous works on the theoretical capabilities of transformers on graph tasks, e.g., in [1] and [2]. But, this is done only in our upper bounds; our lower bounds apply even under these assumptions, thus they also apply to practical settings and demonstrate several expressiveness limitations of the transformer architecture itself.
>
> Questions:
>
> 1) Although it is difficult to exactly formalize property detection tasks on molecules (or other practical tasks on real datasets), we believe they inherently require solving a combinatorial problem on the graph itself. For example, detecting sub-structures that manifest the property, or identifying other combinatorial properties of the graph, such as whether it is connected, contains cycles of certain types, etc. Also, studying such tasks reveals inherent limitations of the architecture itself, which apply to any possible task.
> 2) For example, if we know a certain molecular property appears in a structure containing 4 atoms, our work provides bounds for the width of the transformer to detect such structures. These bounds are, of course, not tight for such tasks, as they may include other properties such as features of each atom, but can provide rough estimates on the required size of the architecture.
> That’s a good suggestion. In the original manuscript, we focused on comparisons only to other transformer architectures, but we will consider adding comparisons also to GNN architectures in the final version.
>
> 3) The intention of this work is to comment on the fundamental capabilities of transformers as their widths and to tightly capture those quantitative trade-offs, and our assumptions (in particular, arbitrary MLPs and positional encodings) are selected with these goals in mind. These assumptions cover a wide range of transformers architectural decisions, and thus, our negative results are particularly expansive. If future work seeks to capture the abilities of transformers to learn these tasks under optimization constraints, we believe it would be possible to do so with more fine-grained constructions that rely on bounded-norm MLPs and standard learnable or sinusoidal positional encodings. If there are particular constraints the reviewer is interested in, we are happy to discuss in more detail.
>
> [1] C. Sanford, D. Hsu, and M. Telgarsky. Transformers, parallel computation, and logarithmic depth, 2024
>
> [2] C. Sanford, D. J. Hsu, and M. Telgarsky. Representational strengths and limitations of transformers, 2024

---

> > ### Comment · Reviewer_hADS · 2025-08-07
> >
> > Thanks for the response. Your clarifications have addressed my concerns, and I can see the contribution of the authors.

---

### Official Review · Reviewer_BMCf · 2025-07-03

**Clarity:** 2
**Significance:** 3
**Originality:** 3
**Rating:** 5
**Confidence:** 3

**Summary:**

The paper characterizes the expressive power of transformers with constant depth and variable width for graph-reasoning tasks. It shows that linear width is sufficient for distinguishing $n$-vertex-cycle graphs from graphs comprised of two $n/2$-cycles, and sufficient and necessary for detecting 2-cycles in directed graphs. For the special case of bounded degree graphs with maximum degree $d$, the paper shows that constant depth and sublinear $O(d \log n)$ width is sufficient to detect 2-cycles. For tasks that require superlinear width, two results are presented: $O(n^{2-1/k})$ width and constant depth is required for subgraph counting for subgraphs of size $k$, and $O(n^2)$ width is necessary for Eulerian cycle verification. Experiments on connectivity and $n$-cycle counting for $n=\{3,4\}$ demonstrate that shallow and wide transformers perform comparable to narrow and deep transformers but have significantly faster training and inference time.

**Questions:**

* If I understand the adjacency encodings correctly, they are not permutation invariant, which might lead to two identical graphs (with differently ordered nodes and thus different adjacency matrices) having different representations. This does not matter for, e.g., subgraph detection, but might have implications for graph-level tasks. Given that the results on the ogbg datasets are quite bad (cf. https://ogb.stanford.edu/docs/leader_graphprop/#ogbg-molhiv for ogbg-molhiv), do you think not being permutation-invariant could be an issue?
* Have you considered/tried to generalize the 1-cycle vs. 2-cycle problem (e.g., considering biconnected components of a restricted graph class such as outerplanar graphs or cactus graphs)? This might strengthen your theoretical contribution.
* What is the "well-accepted" one-cycle conjecture?
* Could you clarify whether subgraphs are induced or not?
* Theorem 5.1: Why is this considered nearly quadratic width (and not quadratic width)?
* Could you provide more details on the following statement: "We used two attention heads to ensure there exists a width for which the model can fit the data for all the evaluated graph sizes". Why do we need two attention heads to fit graphs of all sizes?
* Experiments: Given your focus on small, and thus mostly molecular, datasets, are there specific assumptions you can leverage for molecular graphs? For instance, molecular datasets are usually planar (or even outerplanar), and thus sparse. It might also make sense to explore the performance for the the bounded-degree case for molecules. What are your thoughts on this?

**Ethical Concerns:**

["NO or VERY MINOR ethics concerns only"]

**Final Justification:**

Given the authors' rebuttal, I decided to increase my score. I still consider the fact that the positional embeddings are not permutation-invariant as a weakness, but the overall contribution is sufficient for acceptance.

**Limitations:**

The theoretical results rely on the (common, but strong) assumption that the multi-layer perceptron is a universal approximator as well as arbitrary positional encodings. And the results appear to be non-uniform, i.e., they only hold for graphs up to a certain fixed size.

**Paper Formatting Concerns:**

No concerns

**Quality:**

3

**Strengths And Weaknesses:**

**Strengths**:  Overall, the paper is well-written, easy to follow, and covers a highly relevant topic: Analyzing the tradeoff between depth and width of transformers is a useful contribution that has the potential to better understand and improve transformer-based machine learning models. Moreover, the paper specifically looks at _graph_ tasks, which have so far only been explored to a limited extent. The experiments offer a promising takeaway for practitioners: For certain tasks, shallow and wide transformers might perform equally well to (narrow and) deep transformers, while being faster to train and at inference time.

**Weaknesses**: One perceived weakness is that the proposed encoding schemes render the resulting transformer architecture non-permutation-invariant. This does not matter for tasks such as subgraph or cycle detection, but might be undesirable for graph-level prediction tasks. It is unclear to me whether this is also part of the reason that the results on the ogbg datasets are far away from state-of-the-art (e.g., Graphormer [2] has >80 ROC-AUC on ogbg-molhiv [3]). It could be useful to consult [1] for the comparison of different positional encodings and their expressive power for graph transformers. Another weakness is that some of the tasks (e.g., the 1-cycle vs. 2-cycle problem and the 2-cycle detection in directed graphs) are quite specific and would strengthen the theoretical contribution if generalized. With respect to clarity and structure, there is some room for improvement. The introduction could potentially be made more concise, which might create space to include an additional experimental result from the appendix in the main text. While the paper is easy to read, sometimes it would benefit from more technical details such as more rigorous preliminaries. At times, the paper is difficult to understand without consulting related work. For instance, it would make the paper more self-contained by stating conjectures that theorems depend on, providing some introduction to massively parallel computation (MPC) schemes and properly introducing assumptions and restrictions on the graphs (undirected or directed, simple or multi, etc.) considered in the different tasks. This also applies to, e.g., the title of section 4.2, which is a bit misleading as the lower bound only holds for 2-cycles in directed graphs.

[1] Black, Mitchell, et al. "Comparing Graph Transformers via Positional Encodings." Forty-first International Conference on Machine Learning.

[2] Ying, Chengxuan, et al. "Do Transformers Really Perform Badly for Graph Representation?." Advances in Neural Information Processing Systems.

[3] https://ogb.stanford.edu/docs/leader_graphprop/#ogbg-molhiv

Please find some minor remarks and typos below.

- Line 87: model -> models
- Line 119: empirical tradeoffs "for" spectral Laplacian-based encodings
- Section 3.1: $m$ not introduced
- 198: typo in "linear width is suffices"
- Last sentence in Theorem 4.1 would benefit from some rephrasing
- Lines 230-231: Consider rewriting this statement, it does not seem fully correct to me. Shouldn't you have to check $A^i$ from $i=1,...,n$ to know about connectivity?
- Line 257: graph problem -> graph problems
- Line 281: such that -> that
- Line 285: subgraph -> subgraphs
- Lines 371-371: "this figure" -> Please consider referring to the actual figure
- Line 563: effect -> affect
- Line 566: typo in "will does"
- Line 567: ReLU network -> ReLU networks; and please consider specifying that $\sigma(z)$ is ReLU/$max(0,z)$
- Line 578: whose embedded -> grammar sounds off
- Line 809: appears exactly two pairs -> appears in exactly two pairs?
- Line 922: Missing period at the end of sentence.
- Fig. 1: Inconsistent capitalization of theorems
- Fig. 4-5 have exactly the same caption; where are the results on the connectivity experiments?
- Inconsistent usage of 1-cycle vs. one-cycle

**Justification for score**: My current score reflects some concerns and unaddressed limitations that I have summarized in **Weaknesses** and **Questions**. I am happy to raise my score if these concerns get addressed.

---

> ### Author Rebuttal · Authors · 2025-07-31
>
> We thank the reviewer for the thorough and positive review.
>
> To answer the questions:
>
> - Indeed, the adjacency positional embeddings are not permutation invariant. We note this is also the case for the edge list positional embeddings, which were previously used to study the connection between transformers and the MPC model (in [1] and [2]). We use this embedding for technical convenience, especially since spectral embedding, although useful in practice, are difficult to analyze for combinatorial problems (such as connectivity, subgraph counting etc.). The experiment presented in Table 1 is not aimed at beating or achieving comparable results to SOTA, but rather to justify, even in a small setting that the adjacency encoding is not worse than other encodings such as edge-list and spectral.
>
> - This is a good point. We can generalize the 1-cycle vs. 2-cycles results to the general problem of detecting whether a graph is connected. This is outlined in lines 184-196 in the paper, and uses the method of linear sketching. We will consider formalizing this and presenting the result as a theorem in the final version.
>
> - The 1-cycle vs. 2-cycles problem is the following: Given a graph which is either (1) One cycle with $n$ nodes; or (2) Two cycles, each with $n/2$ nodes, determine which of the two options it is. This problem can be solved by a distributed algorithm in the MPC model by having at most $O(\log(n))$ communication rounds. The 1-cycle vs. 2-cycles conjecture states that it cannot be solved by less than $O(\log(n)) rounds. This is the most commonly assumed conjecture in the context of MPC; see also the Wikipedia page called “1-vs-2 cycles problem” (due to the new NeurIPS policy, we don’t add links in the rebuttal). Many lower bounds and hardness results in the context of MPC rely on this conjecture, and to translate these hardness results, we assume it as well. In the final version, we will add a further discussion and formal definition of this conjecture.
>
> - In Section 5.2 we consider non-induced subgraphs. In [3] the authors distinguish between the two cases, but that is a paper focused entirely on sub-graph counting. Since this is not the main focus of our paper, we focused only on one case, although it may be interesting to study induced sub-graph counting since the results there may differ.
>
> - In the literature of distributed computing (i.e., related to MPC) if the lower bound applies in the case of $n^{2-\epsilon}$ for any $\epsilon > 0$, it is considered only nearly quadratic, rather than quadratic. It is more a matter of terminology.
>
> - As in these experiments, we fix the width and depth of the network in each configuration, we observed that some settings failed to converge with just one attention head. To ensure a fair comparison we wished to fix the number of attention heads, and therefore we used two attention heads across all settings as this ensured convergence.  We will make sure to clarify this in the camera-ready version.
>
> - That’s a good point, and we believe there are. Assuming that the graph is sparse, the adjacency encoding is rather superfluous. It may be replaced by a smaller-dimensional embedding, for example, by projecting each adjacency row using a random or specially crafted projection. This is also the case of bounded-degree graphs, where the adjacency matrix is sparse. In our work, we haven’t specifically tested the effect of width for sparse graphs, but rather provide bounds for general graphs (except for Section 4.3, which focuses on sparse graphs and the 2-cycle detection problem). It is a very interesting future direction to study such bounds on sparse graphs, which are commonly encountered in practice, and whether they improve on the general case, which is explored in our paper.
>
> Regarding the limitations:
>
> Our work indeed assumes universal approximation of the MLP part; this is also done in previous works studying the expressiveness of transformers on graph tasks, e.g., in [1] and [2]. The reason for this is to align with the distributed computing literature, which takes into account global communication rounds, but not local computations. It will be interesting to do a more fine-grained analysis in the case of whether the MLPs are bounded, and how it affects the different bounds. We conjecture that in this case, we would face circuit complexity bounds, which are studied, for example, in [4].
>
> Regarding non-uniformity, this is a general limitation of computational complexity problems. Namely, it is not possible to construct a transformer with constant size (depth, width and bit-precision) that can solve connectivity, sub-graph counting, or other similar combinatorial problems on graphs of any size. Even counting the degree of a single node would require, in the general case, constructing a transformer that counts up to $n$, which requires at least $log(n)$ bits.
>
> Regarding the weaknesses:
>
> About the non-premutation-invariant encodings, it is the same answer as above in the “questions” section. Regarding the paper “Comparing Graph Transformers via Positional Encodings", this is indeed useful, and we will add a citation. We admit we haven’t made a thorough experimental study on different graph embeddings, but rather made an experiment to justify the use of the adjacency encoding we used.
> About the presentation, thank you for the remark. We will try to make the introduction more precise. Also, in the final version, we will add a more thorough overview of the MPC model, the assumption we use (e.g. 1-cycle vs. 2-cycles), and be clearer about the restrictions we make on graphs. We will also change the title of Section 4.2 to make it more precise.
>
> Thank you for carefully reading our paper and spotting the typos, they will be fixed in the final version. We will rewrite the statement in lines 230-231, to check all the powers of the adjacency. Figures 4 and 5 contain a typo in caption; Figure 4 presents the results for 50-nodes on the connectivity task.

---

> > ### Comment · Reviewer_BMCf · 2025-08-04
> >
> > Thank you for your answers!
> >
> > I think you forgot to link the references, could you please specify what papers you refer to with [1]-[4]?
> >
> > And one follow-up question: Do you have any ideas why the experiments failed to converge with only one attention head? Did they only fail for certain widths/depths?

---

> ### Author Response · Authors · 2025-08-04
>
> We apologize for the missing references, and thank you for the additional question. The references are:
>
> [1] C. Sanford, D. Hsu, and M. Telgarsky. Transformers, parallel computation, and logarithmic depth, 2024
>
> [2] C. Sanford, D. J. Hsu, and M. Telgarsky. Representational strengths and limitations of transformers, 2024
>
> [3] Z. Chen, L. Chen, S. Villar, and J. Bruna. Can graph neural networks count substructures?, 2020
>
> [4] W. Merrill, William, and A. Sabharwal. A little depth goes a long way: The expressive power of log-depth transformers, 2025
> ‏
>
> Regarding the question: The experiment in Section 6.1 (Figure 2) tested different depth/width combinations. With one head, the experiment converged only if the width was larger than 63 (the middle column), even though the depth was large. With two heads, it converged for all the depth and width combinations that we tested. Since the goal of this experiment is to test the convergence time, we used the two-heads setting, where all experiments converged, to ensure a fair comparison. Regarding the experiment in Section 6.2, we employed a similar two-head setting to maintain consistency with the previous experiment.
>
> As to why two heads *practically* work better than a single head, we believe this is quite interesting and underexplored, and probably goes beyond the context of graph tasks and the scope of our paper. We conjecture that it is related to optimization issues rather than expressive capabilities. In the original "Attention is all you need" paper, the authors explicitly state that they found multi-head attention to be more beneficial than single-head attention, but they don't provide a reason for that, see Section 3.2.2 there. Since then, using multi-head attention is the common practice, but the theoretical reason for that remains a mystery as far as we know.

---

> > ### Comment · Reviewer_BMCf · 2025-08-08
> >
> > Thanks for the detailed answer! Overall, I think this paper is a valuable contribution and I am leaning towards increasing my score. I will wait for the end of the discussion phase to update the score (and the required 'final justification') in case there is any additional discussion between reviewers and/or AC that might affect my rating.

---

### Decision · Program_Chairs · 2025-09-17

**Decision:**

Accept (spotlight)

**Comment:**

This paper studies the depth–width tradeoffs of transformers on graph algorithmic tasks. The key contributions are: (1) theoretical results showing that constant-depth, linear-width transformers can solve tasks such as connectivity and cycle detection, (2) theoretical results implying that more complex problems, including subgraph counting and Eulerian cycle verification, require nearly quadratic or quadratic width, and (3) empirical evidence that shallow, wide transformers can match the performance of deeper models while offering efficiency benefits in training and inference. The reviewers agreed that the paper provides novel and intellectually interesting insights into the expressiveness of transformers, and that both the theoretical and empirical findings can add value to the community. Notably, the new analysis angle explored by this work leads to novel results and could inspire future works in graph machine learning. I thus recommed a spotlight presentation.

For the camera-ready version, I encourage the authors to (1) better clarify the implications (e.g., advantages and disadvantages) of using non-permutation-invariant encodings, particularly for real-world prediction tasks, (2) improve clarity in the description of graph tasks and encodings (e.g., adjacency rows versus edge lists) as well as their potential limitations, and (3) strengthen the empirical evaluation, for example by providing further analysis of the performance gap on real datasets.